# UGround: Towards Unified Visual Grounding with Unrolled Transformers

**Rui Qian**[1]  **Xin Yin**[2]  **Chuanhang Deng**[1 3]  **Zhiyuan Peng**[4]  **Jian Xiong**[1]  **Wei Zhai**[1]  **Dejing Dou**[1 3]

## Abstract

We present UGround, a **U**nified visual **Ground**ing paradigm that dynamically selects intermediate layers across **U**nrolled transformers as "mask as prompt," diverging from the prevailing pipeline that leverages the fixed last hidden layer as "`<SEG>` as prompt." UGround addresses two primary challenges posed by the prevailing paradigm: (1) its reliance on the fixed last hidden layer, which sequentially amplifies cumulative errors arising from layer-by-layer propagation without intermediate correction, and (2) its use of `<SEG>` as a prompt, which implicitly projects textual embeddings into visual space without explicit spatial cues (*e.g.*, coordinates). Central to UGround is Policy-Prompted Masking, which comprises two key components: Stochastic Skip Connection (SSC) and Mask as Prompt (MasP). SSC is a reinforcement learning policy that, via stochastic sampling, allows each `<SEG>` token to slide across unrolled transformer layers, enabling dynamic layer selection at which it connects to the vision model (*e.g.*, SAM) in a skip-connection fashion. Given the selected hidden layer, MasP uses the similarity map derived from the `<SEG>` token and image tokens as a soft logit mask to prompt SAM for mask generation, offering explicit spatial cues through its activation regions. To validate the effectiveness of UGround, we, for the first time, have unified visual grounding within a single framework from an attribute perspective, spanning from traditional refer expression segmentation to newly proposed reasoning segmentation, single-target to multi-target, positive query to false premise (empty target). All code and models are publicly available at https://github.com/rui-qian/UGround.

[1]College of Computer Science and Artificial Intelligence, Fudan University [2]Zhejiang University [3]BEDI Cloud [4]Shanghai Jiao Tong University. {qiianruii, dengch2000}@gmail.com, xyin@zju.edu.cn, pzy2000@sjtu.edu.cn. Correspondence to: Dejing Dou <dejingdou@gmail.com>.

*Proceedings of the 43ʳᵈ International Conference on Machine Learning*, Seoul, South Korea. PMLR 306, 2026. Copyright 2026 by the author(s).

## 1. Introduction

Visual grounding aims to align referring expressions with their corresponding regions in an image or video (Mao et al., 2016; Yu et al., 2016). Despite existing efforts (Zou et al., 2023a; Liu et al., 2024b; Zou et al., 2023b; Liang et al., 2023; Liu et al., 2023; Wang et al., 2022), it remains largely unexplored under the newly emerging task settings (Lai et al., 2024; Wu et al., 2024; Xia et al., 2024; Rasheed et al., 2024; Ren et al., 2024; Qian et al., 2025). As shown in Table 1, the shifts across recent tasks settings reveal how visual grounding has evolved, from referring to objects by an explicit mention (RES (Rasheed et al., 2024; Ren et al., 2024)) to reasoning over descriptive language in an implicit fashion (RS (Lai et al., 2024; Qian et al., 2026)); from processing single-target queries per instance to handling multi-target scenarios (gRES (Xia et al., 2024), Multi-RS (Ren et al., 2024)); and from responding solely to positive queries to rejecting false premises with empty targets (FP-RES (Wu et al., 2024; Xia et al., 2024)). Such shifts reflect the intrinsic properties of tasks, which we term "attribute variation."

*Table 1.* Attribute variation w.r.t. visual grounding. RES: Referring Expression Segmentation, RS: Reasoning Segmentation, FP: False Premise, gRES: Generalized RES, Multi-RS: Multi-target RS.

| Method | RES | RS | FP-RES | gRES | Multi-RS |
|---|---|---|---|---|---|
| LISA (Lai et al., 2024) | ✓ | ✓ | ✗ | ✗ | ✗ |
| SESAME (Wu et al., 2024) | ✓ | ✓ | ✓ | ✗ | ✗ |
| READ (Qian et al., 2025) | ✓ | ✓ | ✓ | ✗ | ✗ |
| GLaMM (Rasheed et al., 2024) | ✓ | ✓ | ✗ | ✓ | ✗ |
| GSVA (Xia et al., 2024) | ✓ | ✓ | ✓ | ✓ | ✗ |
| PixelLM (Ren et al., 2024) | ✓ | ✓ | ✗ | ✓ | ✓ |
| UGround (Ours) | ✓ | ✓ | ✓ | ✓ | ✓ |

However, we observe that little research has looked into unifying existing tasks within a single framework from an attribute perspective, diverging from prior works (Ren et al., 2024; Rasheed et al., 2024) that primarily focus on versatile capabilities. Generally, perception systems engaging with real-world scenarios are expected to reason over implicit instructions (cognition), interact with one or multiple targets (generality), and appropriately reject queries when necessary (safety). Built upon attribute basis described above, versatility-level extensions, such as grounding in videos (Wei et al., 2025) or conversations (Rasheed et al., 2024), find their footing. This leads us to ask: *Can we design a unified architecture to bridge this gap?*

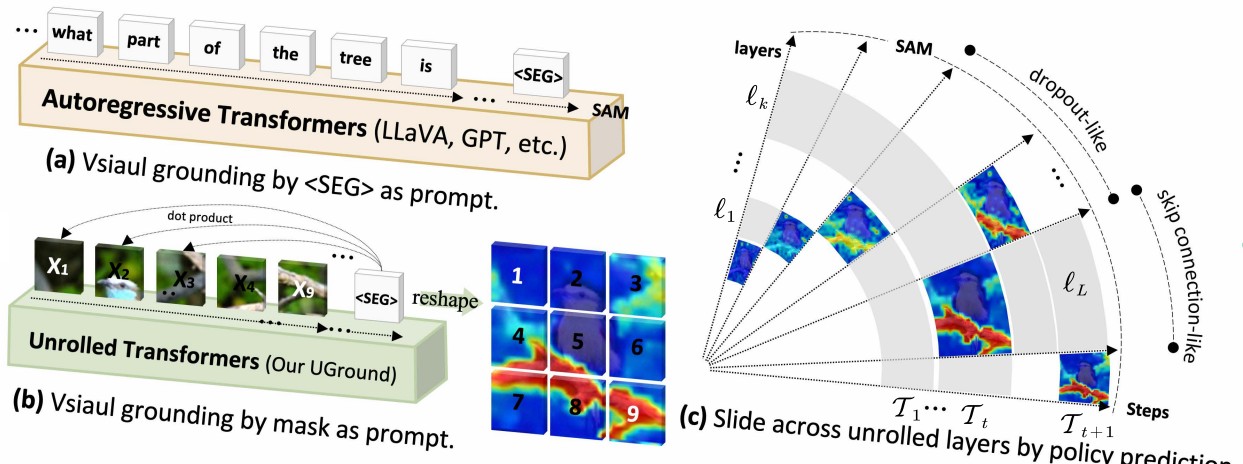

*Figure 1.* Prior works typically use `<SEG>` token embeddings from the last hidden layer as the prompt in (a). In contrast, we leverage the similarity map in Eq. (6), generated from `<SEG>` and image token embeddings across dynamically selected transformer layers, as the prompt in (b). Dynamic layer selection allows the similarity map to slide across transformer layers in (c). Due to the sequential nature of transformers, $\ell_L - \ell_k$ layers are skipped, enabling a direct connection with SAM in a single forward step $\mathcal{T}_t$ ("skip-connection-like"). Across multiple forward steps $\mathcal{T}_1$ to $\mathcal{T}_T$, the connectivity to SAM varies , with only one path activated at each step ("dropout-like").

Bearing this in mind, we revisit the methods of visual grounding presented in Table 1, where we identify two key challenges of the off-the-shelf models: (1) its reliance on the fixed last hidden layer, and (2) its use of `<SEG>` as a prompt. Current Large Multimodal Models (LMMs) based visual grounding typically leverage `<SEG>` token stemming from the fixed last hidden layer of the vision-language model's stacked transformers (*e.g.*, LLaVA) to prompt the downstream vision model (*e.g.*, SAM) as shown in Fig. 1(a). These stacked transformers can be deep, *e.g.*, 32 layers for LLaVA 7B, 40 layers for LLaVA 13B. In this paradigm, only the last hidden layer (layer 40) interacts with the vision SAM directly, whereas the other layers (layers 1-39) are separated. Consider telephone game by analogy, a message is passed from person to person (layer to layer) and becomes increasingly distorted before reaching the final participant (SAM). With only a single path, it sequentially amplifies cumulative errors arising from layer-by-layer propagation without any intermediate correction. In contrast to the similarity map with activation regions in Fig. 1(b), `<SEG>` token, a text placeholder, lacks explicit spatial cues on its own, *e.g.*, coordinates. It works via the alignment from text vocabulary into vision space in an implicit way of a fully connected layer. Inspired by the above findings, a natural idea is whether we can "cheat" this telephone game by letting the final participant (SAM) tap into intermediate layers (*e.g.*, 1–39) in advance, through "mask as prompt."

To this end, we present UGround, a **U**nified visual **Ground**ing paradigm that dynamically selects intermediate layers across **U**nrolled transformers as "mask as prompt." Central to our UGround is Policy-Prompted Masking (PPM), which consists of two key components: Stochastic Skip Connections (SSC) and Mask as Prompt (MasP). As shown in Fig. 1(c), SSC is a stochastic reinforcement learning pol-

icy that allows each `<SEG>` token to slide across unrolled transformer layers, reinforcing dynamic layer selection at which it connects to the vision model (*e.g.*, SAM) in a skip-connection fashion. Remarkably, SSC enables intermediate layers to skip subsequent ones and connect directly with SAM within each forward $\mathcal{T}_t$ ("skip-connection-like"). Across forward $\mathcal{T}_1$ to $\mathcal{T}_T$, such mechanism virtually connects all layers to SAM while activating only one pass at a time ("dropout-like"). Given the selected hidden layer, MasP uses the similarity map derived from the `<SEG>` token and image tokens as a soft logit mask to prompt SAM for mask generation, offering explicit spatial cues through its activation regions in the forward. Particularly, the similarity map is differentiable, beyond its implicit optimization via SAM backpropagation, we further impose a cross-entropy loss against the ground-truth mask to explicitly guide spatial cues in the backward. Our contributions are threefold:

- We have "unified" visual grounding within a single framework from an attribute perspective. Previously, existing works either focus on explicit rather than implicit expressions, or handle single-target instead of multi-target scenarios, or respond solely to positive queries without rejecting false premises (empty target).

- We have "unrolled" stacked transformers, letting vision decoder tap into intermediate layers in a stochastic skip-connection fashion, in the way of "mask as prompt." Importantly, we explicitly supervise the prompted mask (similarity map) against the ground-truth mask to further guide the model on where to attend.

- We conduct extensive experiments on the recently proposed ReasonSeg, the classic RefCOCO(+/g), and gRefCOCO(+/g) dataset. UGround outperforms the state of the art, achieving up to 9.0% gains in cIoU on ReasonSeg and 12.1% gains in N-acc on gRefCOCO(+/g) val sets.

## 2. Related Work

**Large Multimodal Models.** Depending on the trade-offs between computational efficiency and performance ceiling, we categorize LMMs into two groups: (1) *Fusion-centric models*, which aim for deeply fine-grained multimodal interaction by projecting visual, textual, and other modalities into a shared high-dimensional space with cross-attention mechanisms (Alayrac et al., 2022; Wang et al., 2023; Pi et al., 2023; Lv et al., 2023; Zhang et al., 2024a). Flamingo (Alayrac et al., 2022) unlocks few-shot visual reasoning by bridging a frozen language model and a vision encoder with gated cross-attention. VisionLLM (Wang et al., 2023) conditions a language model on image features via cross-attention to handle diverse vision-centric tasks, generating task-specific tokens for open-ended reasoning. GPT4RoI (Zhang et al., 2024a) enables region-of-interest (RoI) reasoning through RoI-aware cross-attention, allowing users to reference any image region in dialogue. Kosmos-2 (Peng et al., 2024) and DetGPT (Pi et al., 2023) equip LMMs with grounding capabilities. (2) *Alignment-centric models*, which employ a divide-and-conquer strategy, projecting the outputs of frozen or lightly fine-tuned unimodal encoders (*e.g.*, ViT for vision, LLaMA/GPT for text) into a shared latent space via a lightweight alignment adapter (*e.g.*, projectors, Q-Formers) (Li et al., 2023; Ye et al., 2023; Li et al., 2025; Liu et al., 2024a; Zhu et al., 2024). Otter (Li et al., 2025) instruction-tunes a Flamingo-style vision–language model on the MIMIC-IT dataset. LLaVA (Liu et al., 2024a) and MiniGPT-4 (Zhu et al., 2024) align a pretrained vision encoder with a frozen LLM using lightweight adapters for open-ended, instruction-following visual reasoning. Unlike LLaVA and MiniGPT-4, which rely on linear projections, BLIP-2 (Li et al., 2023) pioneers the Q-Former to distill visual features into compact queries for alignment with a frozen LLM. mPLUG-OWL (Ye et al., 2023) utilizes a visual abstractor to summarize global image features into condensed semantic tokens. In contrast, our UGround builds upon LLaVA (Liu et al., 2024a) for visual autoregressive modeling and reasoning.

**Unified Segmentation Models.** Depending on the semantic granularity of joint tasks, we divide literature into two groups: *Versatility-oriented segmentation*, which aims to unify multiple functional properties within a single model (Wei et al., 2025; Cheng et al., 2022; Li et al., 2024; Zhang et al., 2024b; Bai et al., 2024; Rasheed et al., 2024), as opposed to prior works dedicated exclusively to semantic (Long et al., 2015; Ronneberger et al., 2015; Badrinarayanan et al., 2017), instance (He et al., 2017; Liu et al., 2018; Carion et al., 2020), or panoptic segmentation (Kirillov et al., 2019b; Xiong et al., 2019; Kirillov et al., 2019a). Mask2Former (Cheng et al., 2022) pioneers a unified approach for image-level semantic, instance, and panoptic segmentation. OMG-Seg (Li et al., 2024) ex-

tends Mask2Former to their video-level counterparts. HyperSeg (Wei et al., 2025) and OMG-LLaVA (Zhang et al., 2024b) distinguish themselves from previous purely visual models (Cheng et al., 2022; Li et al., 2024) by leveraging LMMs to generate interactive text. VideoLISA (Bai et al., 2024) and HyperSeg (Wei et al., 2025) inject complex reasoning capabilities into generic video segmentation models. (2) *Attribute-oriented segmentation*, which seeks to unify the intrinsic property variations within the task itself, spanning from explicit expressions to implicit instructions, one target to many targets, and positive queries to false premises. LISA (Lai et al., 2024) elevates RES to RS by leveraging self-reasoning capabilities of LMMs. SESAME (Wu et al., 2024) and READ (Qian et al., 2025) extend LISA by rejecting false premises. GSVA (Xia et al., 2024) and PixelLM (Ren et al., 2024) generalize RES to gRES by supporting multiple targets. Closest to our work, PixelLM (Ren et al., 2024) cannot tackle empty targets, while GSVA (Xia et al., 2024) fails to handle multi-target reasoning. Attribute-oriented unification is more fundamental, as it can be applied from a versatile perspective to any specialized task. Yet, we observe that little research has looked into unifying existing tasks within a single framework from an attribute perspective, we therefore present UGround.

## 3. Reflection on Visual Grounding

In this section, we first revisit visual grounding with LMMs and then analyze the underlying challenges posed by the typical paradigm presented in (Lai et al., 2024; Wu et al., 2024; Qian et al., 2025).

### 3.1. Revisiting

**Problem Definition:** Let $\mathbf{x}_{img} \in \mathbb{R}^{h \times w \times c}$ denote an input image with height $h$, width $w$, and $c$ channels, and let $\mathbf{x}_{txt}$ represent the paired textual input, which can span from an explicit mention (*e.g.*, "twigs") to an implicit description (*e.g.*, "what part of the tree"). For segmentation, visual grounding aims to generate a mask that localizes the region of $\mathbf{x}_{img}$ described by the textual query $\mathbf{x}_{txt}$ as

$$\Theta_{\mathrm{MLE}} = \arg\max_{\Theta} \mathbb{E}_{\hat{\mathbf{M}} \sim \mathcal{G}_{\theta}(\cdot|\mathbf{x})} \big[ \log \mathcal{G}_{\theta}(\hat{\mathbf{M}} \mid \mathbf{x}) \big], \quad (1)$$

where $\mathbf{x}$ denotes a multimodal sequence of discrete tokens derived from $\mathbf{x}_{img}$ and $\mathbf{x}_{txt}$. $\hat{\mathbf{M}} \in \{0,1\}^{h \times w}$, 1 indicates the foreground and 0 otherwise. $\mathcal{G}_{\theta}$ indicates LMMs-based segmentation, including a multi-modal LLM $\mathcal{G}_{\mathcal{T}}$ and a visual backbone model $\mathcal{G}_{\mathcal{V}}$. Formally, $\mathcal{G}_{\theta} = \mathcal{G}_{\mathcal{V}} \circ \mathcal{G}_{\mathcal{T}}$, $\circ$ denotes the composition operation. As illustrated in Fig. 3, we use LLaVA (Liu et al., 2024a) for $\mathcal{G}_{\mathcal{T}}$ and SAM (Kirillov et al., 2023) for $\mathcal{G}_{\mathcal{V}}$, respectively.

To equip $\mathcal{G}_{\theta}$ with reasoning capabilities of LMMs, LISA (Lai et al., 2024) augments the text vocabulary of

$\mathcal{G}_{\mathcal{T}}$ with a `<SEG>` token. Consider an input sequence $\mathbf{x} = (x_1, x_2, \ldots, x_T)$, $\mathcal{G}_{\mathcal{T}}$ consumes $\mathbf{x}$ as input with $L$ stacked transformer layers, which in turn response hidden states as

$$\mathcal{H}^{(\ell)} = \mathcal{G}_{\mathcal{T}}^{(\ell)}\big(\mathcal{H}^{(\ell-1)}\big), \quad \ell = 1, 2, \ldots, L, \qquad (2)$$

where $\mathcal{H}^{(0)}$ denotes the embeddings of the input sequence $\mathbf{x}$ at the embedding layer. The hidden state at the $\ell$-th layer can be presented as $\mathcal{H}^{(\ell)} = \left\{ \boldsymbol{h}_1^{(\ell)}, \boldsymbol{h}_2^{(\ell)}, \ldots, \boldsymbol{h}_T^{(\ell)} | \boldsymbol{h}_t^{(\ell)} \in \mathbb{R}^d \right\}$. Let $t^*$ be the position of the `<SEG>` token in the input sequence $\mathbf{x}$, then its corresponding hidden state at the $\ell$-th layer is $\boldsymbol{h}_{t^*}^{(\ell)}$, which is finally fed into $\mathcal{G}_{\mathcal{V}}^{dec}$ to generate the binary segmentation mask $\hat{\mathbf{M}}$ as

$$\hat{\mathbf{M}} = \mathcal{G}_{\mathcal{V}}^{dec}(\mathbf{f}, \boldsymbol{h}_{seg}), \quad \boldsymbol{h}_{seg} = \varphi(\boldsymbol{h}_{t^*}^{(\ell)}) \in \mathbb{R}^d, \qquad (3)$$

where $\varphi(\cdot)$ is a multilayer perceptron (MLP) projection layer, $\mathbf{f}$ is the visual features of the input image $\mathbf{x}_{img}$ extracted by the vision backbone $\mathcal{G}_{\mathcal{V}}^{enc}$ in SAM (Kirillov et al., 2023). However, we observe that existing works (Lai et al., 2024; Wu et al., 2024; Xia et al., 2024; Rasheed et al., 2024; Ren et al., 2024; Qian et al., 2025) predominantly utilize the embedding from the fixed last hidden layer ($\ell = L$) as the `<SEG>` token representation $\boldsymbol{h}_{seg}$, while the impact of intermediate layers remains largely unexplored.

### 3.2. Analysis

**Why Dynamic Layer Selection.** To analyze the impact of intermediate layers, we train UGround on the ReasonSeg *train* set with two strategies: (1) relying solely on the last hidden layer ($\ell = L$), and (2) dynamically selecting from the intermediate layers ($\ell \in \{1, \ldots, L-1\}$). Qualitatively, we visualize the similarity maps across layers 1–40 of $\mathcal{G}_{\mathcal{T}}$ in Fig. 5. We observe that the fixed-last-layer strategy produces noisy similarity maps in intermediate layers (Fig. 5(a)), whereas the intermediate-layer counterpart yields a more discriminative foreground (Fig. 5(b)). Quantitatively, we plot the predicted cIoU of the embeddings across layers 1–40. Fig.2(a) shows that every layer from 10–40 improves over the fixed-last-layer strategy by a clear and consistent margin. We further plot the per-layer similarity-map loss with respect to the ground-truth mask. Fig.2(b) shows that the intermediate-layer strategy begins converging from layer 19 and ultimately reaches a lower loss level, while the fixed-last-layer strategy only starts converging from layer 28, indicating that dynamic layer selection accelerates convergence. Dynamic layer selection provides insight into the behavior of intermediate layers. By allowing these layers to directly participate in interactions and joint training with SAM, akin to "cheating" in a phone game (*i.e.*, receiving intermediate hints), each layer's performance is improved, ultimately resulting in an overall enhancement.

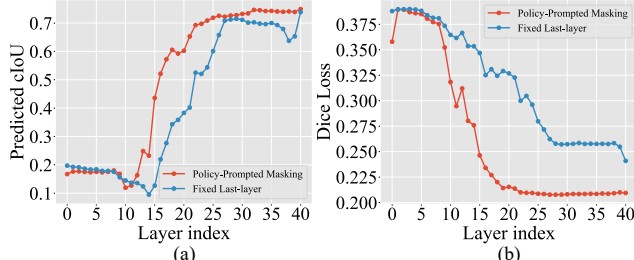

*Figure 2.* (a) shows that the dynamic layer selection strategy outperforms the fixed last hidden layer strategy across all middle layers. (b) plots the loss of similarity maps against the soft ground-truth mask, demonstrating faster convergence at intermediate layers.

**Why Similarity as Mask.** To investigate whether the similarity map can serve as a prompt, we conduct two analyses. First, to measure the consistency ($\mathcal{M}_{\mathrm{IoU}}$) between the similarity map and the ground-truth mask, we compute the grid search-based Intersection over Union (IoU) following READ (Qian et al., 2025). Second, to probe whether SAM can understand the semantics encoded in the similarity map, we directly use the similarity map as a logit mask to prompt the original SAM (Kirillov et al., 2023) (without any additional training), denoted by $\mathcal{M}_{\mathrm{prompt}}$. Table 2 shows that SAM can potentially leverage the similarity map as a prompt (17.0% vs. 48.4%). Further, when the similarity map is directly converted into a binary mask for prediction, its performance measured by $\mathcal{M}_{\mathrm{IoU}}$ can even rival that of the trained `<SEG>`$_{\mathrm{prompt}}$, as evidenced by a 4.3% higher cIoU on the ReasonSeg *test* set (35.0% vs. 30.7%) in the 3$^{\mathrm{rd}}$ row.

*Table 2.* Quantitative analysis of the similarity maps on the ReasonSeg *test* split. $\mathcal{M}_{\mathrm{prompt}}$ indicates mask as a prompt for the original SAM, while $\mathcal{M}_{\mathrm{IoU}}$ quantifies the overlap between the similarity map and the ground-truth mask. `<SEG>`$_{\mathrm{prompt}}$ refers to `<SEG>` token as prompt for SAM.

| Method | $\mathcal{M}_{\mathrm{prompt}}$ | | $\mathcal{M}_{\mathrm{IoU}}$ | | `<SEG>`$_{\mathrm{prompt}}$ | |
|---|---|---|---|---|---|---|
| | gIoU | cIoU | gIoU | cIoU | gIoU | cIoU |
| LISA-7B (Lai et al., 2024) | 16.1 | 17.0 | 32.2 | 35.6 | 47.3 | 48.4 |
| GSVA-7B (Xia et al., 2024) | 15.0 | 14.7 | 33.2 | 37.4 | 42.8 | 43.8 |
| SESAME-7B (Wu et al., 2024) | 15.6 | 15.6 | 31.6 | 35.0 | 34.9 | 30.7 |

**Summary**. Analysis of the similarity maps reveals the following: (a) Similarity maps in LMMs are highly consistent with the ground-truth mask, suggesting that they can potentially serve as prompts for generating the segmentation mask. (b) Dynamic layer selection allows the intermediate layers to directly interact with SAM, resulting in a more discriminative representation, as reflected by the activated regions in the similarity maps. Motivated by the insights from the similarity maps, we leverage it in three ways: as a logit mask to prompt SAM (prompt), as a loss constraint to guide the model on where to "attend" (constraint), and as a policy reward for dynamic layer selection (signal) (see Appendix D for further analysis).

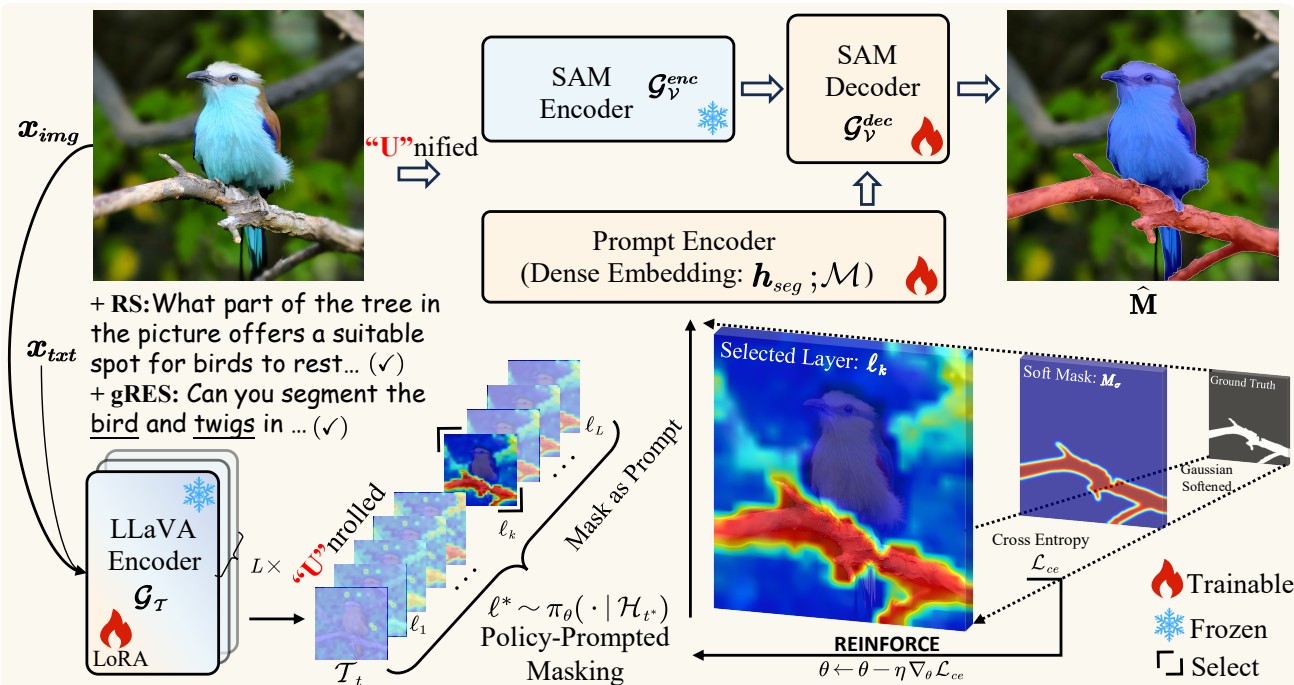

*Figure 3.* Overview of our proposed UGround. Central to UGround is Policy-Prompted Masking (PPM), which stochastically selects layer $\ell^*$ among "**U**nrolled" transformers from a policy distribution $\pi_\theta(\cdot \mid \mathcal{H}_{t^*})$ at which it connects to the vision model (*e.g.*, SAM) in a skip-connection fashion. Given the layer $\ell^*$, MasP uses the similarity map as a soft logit mask to prompt SAM for mask generation, wherein we advance visual grounding within a "**U**nified" paradigm from an attribute perspective.

## 4. Proposed UGround

In this section, we present UGround, which dynamically selects intermediate layers across unrolled transformers and connects them to SAM in a skip-connection fashion, where the similarity map serves as a prompt, thereby contributing to a unified visual grounding paradigm. As depicted in Fig. 3, central to UGround is Policy-Prompted Masking (PPM), consisting of two components: Stochastic Skip Connection (SSC) and Mask as Prompt (MasP). SSC is a reinforcement learning policy that stochastically allows each <SEG> token to slide across unrolled transformer layers, dynamically selecting where it connects to the vision model (*e.g.*, SAM) via a skip connection. MasP, given the selected layer, leverages the similarity map between the <SEG> token and image tokens as a soft logit mask to prompt SAM, providing explicit spatial cues through its activation regions. As the key novelty of our work is PPM, we present it first in Sec. 4.1, then the training objectives in Sec. 4.2.

### 4.1. Policy-Prompted Masking

**Stochastic Skip Connection.** Prior works (Lai et al., 2024; Wu et al., 2024; Xia et al., 2024; Rasheed et al., 2024; Ren et al., 2024; Qian et al., 2025) empirically use the embedding from the fixed last hidden layer ($\ell = L$) as the <SEG> token representation $h_{seg}$. In contrast, we exploit intermediate layers ($\ell \in \{1, \ldots, L-1\}$) to derive more discriminative semantics in a skip-connection fashion. To this end, we formu-

late the problem of selecting a layer for the <SEG> token as a reinforcement learning task. Each layer selection is treated as an action $a \equiv \ell^*$ taken in the state $s \equiv h_{t^*}$, which corresponds to the unrolled <SEG> embeddings across layers. The agent receives a reward $r$ that reflects the consistency of the similarity map with the ground-truth mask. Specifically, let $\mathcal{H}_{t^*}$ represent the hidden states of the <SEG> token from all layers ($\ell \in \{1, \ldots, L\}$) at position $t^*$ of the input sequence $\mathbf{x}$ as $\mathcal{H}_{t^*} = \{h_{t^*}^{(1)}, h_{t^*}^{(2)}, \ldots, h_{t^*}^{(L)}\} \in \mathbb{R}^{L \times d}$, let $\mathbf{w} \in \mathbb{R}^{L \times d}$ denote learnable weights. The policy distribution $\pi_\theta(\cdot \mid \mathcal{H}_{t^*})$ is defined as

$$\pi_\theta(\ell \mid \mathcal{H}_{t^*}) = \frac{\exp(s_\ell)}{\sum_{j=1}^{L} \exp(s_j)}, s_\ell = h_{t^*}^{(\ell)} \cdot \mathbf{w}_\ell, \quad (4)$$

where $s_\ell$ is the logit scores for each layer. $h_{t^*}^{(\ell)}$ is the <SEG> embedding at layer $\ell$ and $\mathbf{w}_\ell$ is the corresponding learnable weight. The agent chooses a layer $\ell$ according to a categorical distribution parameterized by learnable weights $\mathbf{w}$. During training, the layer $\ell^*$ is sampled from this distribution to allow exploration as $\ell^* \sim \pi_\theta(\cdot \mid \mathcal{H}_{t^*})$, To reduce variance, we introduce a baseline $b$, which can be an EMA-smoothed reward or a critic network prediction. The actual reward $r$ is derived from the similarity map, as described in Algorithm (1) steps 5-8. We therefore formulate the REINFORCE gradient as

$$\nabla_\theta J(\theta) = \mathbb{E}_{\ell^* \sim \pi_\theta(\cdot \mid \mathcal{H}_{t^*})} \Big[ (r - b) \nabla_\theta \log \pi_\theta(\ell^* \mid \mathcal{H}_{t^*}) \Big], \quad (5)$$

where the log-probability of the sampled action $a \equiv \ell^*$ is weighted by the advantage $r - b$. To get the reward $r$, we first compute the similarity map $\mathcal{M}$ between the <SEG> token and image tokens at layer $\ell$ and its corresponding soft mask $\mathcal{M}_\sigma$. Consider a sequence of discrete image tokens $\mathbf{z} = (z_1, z_2, \ldots, z_k)$ in the input sequence $\mathbf{x}$, where $k$ is the number of image tokens. The image token embeddings across layers are then denoted as $\mathcal{H}_{\mathbf{z}} = \{\boldsymbol{h}_z^{(\ell)} \mid z \in \mathbf{z}, \ell = 1, \ldots, L\} \in \mathbb{R}^{k \times L \times d}$, where $\boldsymbol{h}_z^{(\ell)}$ is the embedding of the $z$-th image token at layer $\ell$. Given the <SEG> token embedding $\boldsymbol{h}_{t^*}^{(\ell)} \in \mathbb{R}^d$ and the image token embeddings $\mathcal{H}_{\mathbf{z}}^{(\ell)} \in \mathbb{R}^{k \times d}$ at the selected layer $\ell$, the similarity scores can be computed as

$$\mathcal{S}^{(\ell)} = \mathcal{H}_{\mathbf{z}}^{(\ell)} \, {h_{t^*}^{(\ell)}}^\top \in \mathbb{R}^k, \tag{6}$$

where each element $\mathcal{S}_i^{(\ell)} = (h_{z_i}^{(\ell)})^\top h_{t^*}^{(\ell)}$ represents the similarity between the <SEG> token and the $i$-th selected image token at layer $\ell$. $\mathcal{S}^{(\ell)}$ are arranged in a 2D grid, and interpolated to obtain the soft logit mask $\mathcal{M} \in [0, 1]^{H \times W}$. Let the ground-truth mask be $M \in \{0, 1\}^{H \times W}$, which we regard as a function $M : \{1, \ldots, H\} \times \{1, \ldots, W\} \to \{0, 1\}$ over the discrete pixel grid. We extend $M$ to the continuous domain $\mathbb{R}^2$ by assigning each pixel a constant value on its corresponding region in the plane. Applying Gaussian smoothing to this extended function yields a continuous-valued function $M_\sigma : \mathbb{R}^2 \to [0, 1]$, where $M_\sigma$ serves as the soft label (heatmap). Formally, $M_\sigma$ can be expressed as a convolution integral as $M_\sigma(u, v) = \iint_{\mathbb{R}^2} M(x, y) \, G_\sigma(u - x, v - y) \, dx \, dy$, where $G_\sigma(x, y) \propto \exp(-\frac{x^2 + y^2}{2\sigma^2})$, is a 2D Gaussian kernel with standard deviation $\sigma$. The reward for REINFORCE is computed using the similarity map $\mathcal{M}$ and the soft label $M_\sigma$ as $r = -\big(\mathcal{L}_{bce}(\mathcal{M}, M_\sigma) + \mathcal{L}_{dice}(\mathcal{M}, M_\sigma)\big)$, where $\mathcal{L}_{bce}$ and $\mathcal{L}_{dice}$ denote the Gaussian BCE and Dice losses defined over the similarity map $\mathcal{M}$. The policy loss is then defined as

$$\mathcal{L}_{\text{policy}} = -(r - b_t) \log \pi_\theta(\ell^* \mid \mathcal{H}_{t^*}). \tag{7}$$

Algorithm (1) outlines a single forward pass of the PPM. The transformer layers are stacked sequentially. By skipping the subsequent $L - \ell^*$ layers, the PPM allows layer $\ell^*$ to connect directly to SAM in a skip-connection manner, thereby mitigating cumulative errors that arise from layer-by-layer propagation through intermediate correction (i.e., a shortcut that bypasses subsequent layers). Across multiple forward passes (from 1 to $\mathcal{T}$), the PPM dynamically alters the model's internal connectivity, analogous to the dropout mechanism: although all layers are virtually connected to SAM, only one path is activated per pass. This stochastic sampling can be interpreted as a form of Monte Carlo uncertainty estimation (Gal & Ghahramani, 2016), which quantifies predictive uncertainty via multiple stochastic forward passes. In the context of PPM, each forward pass activates a different layer-path to SAM, and the ensemble

---

**Algorithm 1** Policy-Prompted Masking Algorithm

**Require:** Given unrolled transformer layers with $L$ layers, $\mathcal{H}_{t^*} \in \mathbb{R}^{L \times d}$ is hidden states of <SEG> token, $\mathcal{H}_{\mathbf{z}} \in \mathbb{R}^{k \times L \times d}$ is hidden states of image tokens; $\mathbf{w}$ is learnable weights $\mathbf{w} \in \mathbb{R}^{L \times d}$; $M_\sigma : \mathbb{R}^2 \to [0, 1]$, where $M_\sigma$ is the soft heatmap, with a Gaussian kernel $G_\sigma(x, y)$; $r$ is the reward score, $b$ is the baseline, $M \in \{0, 1\}^{H \times W}$ is ground-truth mask; $\mathcal{T}$ denotes the number of forward passes, each of which corresponds to an independent policy sampling followed by a mask prediction.

**Ensure:** Selected layer $\ell_t^*$ and similarity map $\mathcal{M}_t$

1: **for** $t = 1, \ldots, \mathcal{T}$ **do**
2:    $s_\ell = \boldsymbol{h}_{t^*}^{(\ell)} \cdot \mathbf{w}_\ell$ for $\ell = 1, \ldots, L$
3:    $\pi_\theta(\ell \mid \mathcal{H}_{t^*}) = \frac{\exp(s_\ell)}{\sum_{j=1}^L \exp(s_j)}$
4:    $\ell^* \sim \pi_\theta(\cdot \mid \mathcal{H}_{t^*})$
5:    $\mathcal{S}^{(\ell^*)} = \mathcal{H}_{\mathbf{z}}^{(\ell^*)} \, {h_{t^*}^{(\ell^*)}}^\top \in \mathbb{R}^k$
6:    $\mathcal{M} \in [0, 1]^{H \times W} \leftarrow \mathcal{S}^{(\ell^*)}$
7:    $M_\sigma(u, v) = \iint_{\mathbb{R}^2} M(x, y) \, G_\sigma(u - x, v - y) \, dx \, dy$
8:    $r = -\big(\mathcal{L}_{bce}(\mathcal{M}, M_\sigma) + \mathcal{L}_{dice}(\mathcal{M}, M_\sigma)\big)$
9:    $b_t = \alpha b_{t-1} + (1 - \alpha) r$
10:   $\mathcal{L}_{\text{policy}} = -(r - b_t) \log \pi_\theta(\ell^* \mid \mathcal{H}_{t^*})$
11:   $\theta \leftarrow \theta - \eta \nabla_\theta \mathcal{L}_{\text{policy}}$
12: **end for**
13: **return** $\{\ell_t^*, \mathcal{M}_t\}_{t=1}^{\mathcal{T}}$

---

of these paths (i.e., 32 in LLaVA) alleviates over-reliance on any single trajectory, enhancing the model's robustness.

**Mask as Prompt.** Compared with <SEG> tokens, the similarity map provides more explicit spatial cues, as analyzed in Sec. 3.2. We use the similarity map as a logit mask to prompt SAM. Accordingly, we reformulate Eq. (3) as $\hat{\mathbf{M}} = \mathcal{G}_{\mathcal{V}}^{dec}(\mathbf{f}, \boldsymbol{h}_{seg}, \mathcal{M})$. The similarity map is continuously differentiable, allowing its gradients to be backpropagated through the SAM module for implicit optimization. To further guide the model to focus on target regions, we supervise the similarity map $\mathcal{M}$ by imposing a constraint (e.g., using binary cross-entropy (BCE) or Dice loss) against a soft ground-truth mask $M_\sigma$, providing explicit guidance on where the model should "attend."

## 4.2. Training Objectives

Apart from the text generation loss $\mathcal{L}_{txt}$ in $\mathcal{G}_{\mathcal{T}}$, and the segmentation mask loss $\mathcal{L}_{mask}$ in $\mathcal{G}_{\mathcal{V}}^{dec}$ (Lai et al., 2024; Wu et al., 2024), we use policy loss $\mathcal{L}_{\text{policy}}$ for stochastic skip connection, and binary cross-entropy (BCE) loss and Dice loss for similarity map as

$$\begin{aligned} \mathcal{L}_{\text{policy}} &= -(r - b_t) \log \pi_\theta(\ell^* \mid \mathcal{H}_{t^*}), \\ \mathcal{L}_{\mathcal{M}} &= \lambda_{bce} \mathcal{L}_{bce}(\mathcal{M}, M_\sigma) + \lambda_{dice} \mathcal{L}_{dice}(\mathcal{M}, M_\sigma), \end{aligned} \tag{8}$$

where $M_\sigma$ is the soft ground-truth target, $\lambda$ denotes the loss weight, $\mathcal{M}$ is the similarity map. The overall objective $\mathcal{L}$ summarizes those losses in Eq. (8), weighted by $\lambda_{txt}$, $\lambda_{mask}$, $\lambda_{\mathcal{M}}$, and $\lambda_{policy}$ as

$$\mathcal{L} = \lambda_{txt} \mathcal{L}_{txt} + \lambda_{mask} \mathcal{L}_{mask} + \lambda_{\mathcal{M}} \mathcal{L}_{\mathcal{M}} + \lambda_{policy} \mathcal{L}_{\text{policy}}. \tag{9}$$

*Table 3.* Benchmarking reasoning segmentation models on the ReasonSeg dataset, sorted in ascending order of cIoU on the test set. *: reproduced from official models. ft: fine-tuned on 239 samples.

| Method | val | | test | | | | | |
| | overall | | short query | | long query | | overall | |
| | gIoU | cIoU | gIoU | cIoU | gIoU | cIoU | gIoU | cIoU |
|---|---|---|---|---|---|---|---|---|
| X-Decoder (Zou et al., 2023a) | 22.6 | 17.9 | 20.4 | 11.6 | 22.2 | 17.5 | 21.7 | 16.3 |
| Grounded-SAM (Liu et al., 2024b) | 26.0 | 14.5 | 17.8 | 10.8 | 22.4 | 18.6 | 21.3 | 16.4 |
| SEEM (Zou et al., 2023b) | 25.5 | 21.2 | 20.1 | 11.5 | 25.6 | 20.8 | 24.3 | 18.7 |
| OVSeg (Liang et al., 2023) | 28.5 | 18.6 | 18.0 | 15.5 | 28.7 | 22.5 | 26.1 | 20.8 |
| GRES (Liu et al., 2023) | 22.4 | 19.9 | 17.6 | 15.0 | 22.6 | 23.8 | 21.3 | 22.0 |
| *SESAME (Wu et al., 2024) | 40.3 | 41.6 | 28.9 | 26.3 | 37.3 | 31.9 | 34.9 | 30.7 |
| LLaVA1.5-7B + OVSeg (Lai et al., 2024) | 38.2 | 23.5 | 24.2 | 18.7 | 44.6 | 37.1 | 39.7 | 31.8 |
| *GSVA (Xia et al., 2024) | 45.6 | 41.5 | 37.9 | 36.5 | 44.3 | 46.0 | 42.8 | 43.8 |
| *PixelLM (Ren et al., 2024) | 49.7 | 49.6 | 39.5 | 38.8 | 49.5 | 45.6 | 47.1 | 44.3 |
| LISA-7B (Lai et al., 2024) | 52.9 | 54.0 | 40.6 | 40.6 | 49.4 | 51.0 | 47.3 | 48.4 |
| HyperSeg-3B (Wei et al., 2025) | 59.2 | 56.7 | - | - | - | - | - | - |
| VISA-7B (Yan et al., 2024) | 52.7 | 57.8 | - | - | - | - | - | - |
| VideoLISA-3.8B (Bai et al., 2024) | 61.4 | 67.1 | 43.8 | 42.7 | 56.9 | 57.7 | 53.8 | 54.4 |
| LISA-7B-LLaVA1.5 (ft) (Lai et al., 2024) | 61.3 | 62.9 | 48.3 | 46.3 | 57.9 | 59.7 | 55.6 | 56.9 |
| READ-7B-LLaVA1.5 (ft) (Qian et al., 2025) | 59.8 | 67.6 | 52.6 | 49.5 | 60.4 | 61.0 | 58.5 | 58.6 |
| LISA++-7B-LLaVA1.5 (ft) (Yang et al., 2023) | 64.2 | 68.1 | 49.6 | **51.1** | 59.3 | 61.7 | 57.0 | 59.5 |
| RSVP-GPT (Lu et al., 2025) | 64.7 | 63.1 | **55.4** | 50.4 | 61.9 | 62.5 | 60.3 | 60.0 |
| UGround-7B-LLaVA1.5 (ft) | **66.1** | **72.1** | 55.1 | 48.5 | **66.3** | **70.2** | **63.6** | **65.4** |
| LLaVA1.5-13B + OVSeg (Lai et al., 2024) | 37.9 | 26.4 | 27.1 | 19.4 | 46.1 | 40.6 | 41.5 | 34.1 |
| LISA-13B-LLaVA1.5 (Lai et al., 2024) | 57.7 | 60.3 | 50.8 | 50.0 | 54.7 | 50.9 | 53.8 | 50.8 |
| LISA-13B-LLaVA1.5(ft) (Lai et al., 2024) | 65.0 | 72.9 | 55.4 | 50.6 | 63.2 | 65.3 | 61.3 | 62.2 |
| READ-13B-LLaVA1.5 (ft) (Qian et al., 2025) | - | - | 55.4 | **53.7** | 64.4 | 65.1 | 62.2 | 62.8 |
| UGround-13B-LLaVA1.5 (ft) | **67.9** | **74.9** | **57.2** | 50.9 | **67.5** | **69.4** | **65.0** | **65.5** |

*Table 4.* Benchmarking referring segmentation models on the RefCOCO(+/g) dataset, sorted in ascending order of cIoU on the RefCOCOg val set.

| Method | RefCOCO | | | RefCOCO+ | | | RefCOCOg | |
| | val | testA | testB | val | testA | testB | val(U) | test(U) |
|---|---|---|---|---|---|---|---|---|
| MCN (Luo et al., 2020) | 62.4 | 64.2 | 59.7 | 50.6 | 55.0 | 44.7 | 49.2 | 49.4 |
| VLT (Ding et al., 2021) | 67.5 | 70.5 | 65.2 | 56.3 | 61.0 | 50.1 | 55.0 | 57.7 |
| CRIS (Wang et al., 2022) | 70.5 | 73.2 | 66.1 | 62.3 | 68.1 | 53.7 | 59.9 | 60.4 |
| LAVT (Yang et al., 2022) | 72.7 | 75.8 | 68.8 | 62.1 | 68.4 | 55.1 | 61.2 | 62.1 |
| X-Decoder (Zou et al., 2023a) | - | - | - | - | - | - | 64.6 | - |
| ReLA (Liu et al., 2023) | 73.8 | 76.5 | 70.2 | 66.0 | 71.0 | 57.7 | 65.0 | 66.0 |
| SEEM (Zou et al., 2023b) | - | - | - | - | - | - | 65.7 | - |
| Segment Anyword (Liu et al., 2025) | 55.3 | 47.9 | 66.0 | 55.6 | 47.4 | 67.0 | 58.4 | 60.1 |
| VISA-7B (Yan et al., 2024) | 72.4 | 75.5 | 68.1 | 59.8 | 64.8 | 53.1 | 65.5 | 66.4 |
| SESAME (Wu et al., 2024) | 74.7 | - | - | 64.9 | - | - | 66.1 | - |
| LISA-7B (Lai et al., 2024) | 74.9 | 79.1 | 72.3 | 65.1 | 70.8 | 58.1 | 67.9 | 70.6 |
| CoRes (Bao et al., 2024) | 76.0 | 78.6 | 72.5 | 65.1 | 70.0 | 58.6 | 69.0 | 70.7 |
| PixelLM-7B (Ren et al., 2024) | 73.0 | 76.5 | 68.2 | 66.3 | 71.7 | 58.3 | 69.3 | 70.5 |
| READ-7B (Qian et al., 2025) | 78.1 | 80.2 | 73.2 | 68.4 | 73.7 | 60.4 | 70.1 | 71.4 |
| SegLLM-7B (Wang et al., 2025) | 80.2 | 81.5 | 75.4 | 70.3 | 73.0 | 62.5 | 72.6 | 73.6 |
| GSVA-7B (Xia et al., 2024) | 77.2 | 78.9 | 73.5 | 65.9 | 69.6 | 59.8 | 72.7 | 73.3 |
| OMG-LLaVA (Zhang et al., 2024b) | 78.0 | 80.3 | 74.1 | 69.1 | 73.1 | 63.0 | 72.9 | 72.9 |
| GLaMM-7B (Rasheed et al., 2024) | 79.5 | 83.2 | 76.9 | 72.6 | **78.7** | 64.6 | 74.2 | 74.9 |
| UGround-7B | **80.6** | **83.5** | **77.7** | **72.8** | 77.5 | **65.6** | **74.7** | **76.1** |

*Table 5.* Benchmarking generalized referring expression segmentation (GRES) models on the gRefCOCO (Liu et al., 2023) dataset, sorted in ascending order of cIoU on the gRefCOCO validation set. Values are taken from (Liu et al., 2023). N-acc.: the accuracy of correct null-target classification. ft: fine-tuned on gRefCOCO training set.

| Method | Validation Set | | | Test Set A | | | Test Set B | | |
| | gIoU | cIoU | N-acc. | gIoU | cIoU | N-acc. | gIoU | cIoU | N-acc. |
|---|---|---|---|---|---|---|---|---|---|
| MattNet (Yu et al., 2018) | 48.24 | 47.51 | 41.15 | 59.30 | 58.66 | 44.04 | 46.14 | 45.33 | 41.32 |
| LTS (Jing et al., 2021) | 52.70 | 52.30 | - | 62.64 | 61.87 | - | 50.42 | 49.96 | - |
| VLT (Ding et al., 2021) | 52.00 | 52.51 | 47.17 | 63.20 | 62.19 | 48.74 | 50.88 | 50.52 | 47.82 |
| CRIS (Wang et al., 2022) | 56.27 | 55.34 | - | 63.42 | 63.82 | - | 51.79 | 51.04 | - |
| LAVT (Yang et al., 2022) | 58.40 | 57.64 | 49.32 | 65.90 | 65.32 | 49.25 | 55.83 | 55.04 | 48.46 |
| ReLA (Liu et al., 2023) | 63.60 | 62.42 | 56.37 | 70.03 | 69.26 | 59.02 | 61.02 | 59.88 | 58.40 |
| LISA-Vicuna-7B (Lai et al., 2024) | 32.21 | 38.72 | 2.71 | 48.54 | 52.55 | 6.37 | 39.65 | 44.79 | 5.00 |
| GSVA-Vicuna-7B (Xia et al., 2024) | 63.32 | 61.70 | 56.45 | 70.11 | 69.23 | 63.50 | 61.34 | 60.26 | 58.42 |
| LISA-Vicuna-7B (ft) (Lai et al., 2024) | 61.63 | 61.76 | 54.67 | 66.27 | 68.50 | 50.01 | 58.84 | 60.63 | 51.91 |
| GSVA-Vicuna-7B (ft) (Xia et al., 2024) | 66.47 | 63.29 | 62.43 | 71.08 | 69.93 | 65.31 | 62.23 | 60.47 | 60.56 |
| UGround-LLaVA1.5-7B(ft) | **72.46** | **65.56** | **74.53** | **74.29** | **70.87** | **73.93** | **66.85** | **61.84** | **71.22** |

# 5. Experiment

## 5.1. Experimental Setting

We observe that the center scaling and cropping operation in CLIP may crop out target objects. To address this issue and to align the similarity map with SAM's input, we adopt SAM's transformation strategy, scaling the longer side of the image to 336 and padding the image for the CLIP input 336×336. For results on ReasonSeg, UGround-7B-LLaVA1.5 is trained on 1 NVIDIA A100 GPU 80GB for about 2 days, UGround-13B-LLaVA1.5 for about 13 hours. The training relies solely on ReasonSeg and a referring segmentation dataset (~10k images). We use LoRA (Hu et al., 2022) for efficient fine-tuning, using $lora\_r = 8$ for 7B and 64 for 13B. We use the AdamW (Loshchilov & Hutter, 2019) optimizer with an initial learning rate of 3e-4, scheduled by WarmupDecayLR with 100 warmup steps. Loss weights are set as $\lambda_{mask} = \lambda_{txt} = 1.0$, $\lambda_{policy} = 0.1$, $\lambda_{bce} = 2.0$, and $\lambda_{dice} = 4.0$.

## 5.2. Comparison with the State-of-the-Art

**Results on ReasonSeg Dataset.** In Table 3, UGround achieves competitive performance compared with the recent state-of-the-art RSVP-GPT. On the ReasonSeg validation set, UGround-7B improves over RSVP-GPT by +1.4% gIoU and +9.0% cIoU. On the test set, UGround surpasses RSVP-GPT by +3.3% gIoU and +5.4% cIoU. At the larger scale, UGround-13B further pushes the state of the art, reaching 67.9% gIoU / 74.9% cIoU on val and 65.0% gIoU / 65.5% cIoU on test, outperforms READ-13B by up to +2.8% gIoU and +2.7% cIoU. These results validate UGround's effectiveness in complex reasoning-oriented segmentation.

**Results on RefCOCO(+/g) Dataset.** In Table 4, UGround-7B achieves stronger overall performance than the recent state-of-the-art GLaMM-7B. On RefCOCO, UGround surpasses GLaMM-7B by +1.1% on val and +0.8% on testB. On RefCOCO+, it exceeds GLaMM-7B by +1.0% on testB and remains competitive on val and testA. On RefCOCOg, UGround reaches 76.1% on test(U), improving +1.2% over GLaMM-7B, highlighting its generalization across classical referring expression segmentation benchmarks.

**Results on gRefCOCO Dataset.** In Table 5, UGround-7B achieves substantially stronger performance compared with the recent state-of-the-art GSVA-7B (ft) on the gRefCOCO dataset. On the validation set, UGround improves over GSVA-7B (ft) by +5.99% gIoU, +2.27% cIoU, and +12.10% N-acc, respectively. On Test A, UGround surpasses GSVA-7B (ft) by +3.21% gIoU, +0.94% cIoU, and +8.62% N-acc. On Test B, UGround also outperforms the previous SOTA across all metrics. These results demonstrate that UGround not only reflects its strength in multi-object segmentation but also excels in correctly identifying empty targets (N-acc).

## 5.3. Ablation Study

In this section, we conduct an ablation study to analyze the contribution of each component. We report the gIoU and cIoU performance on the *val* set of ReasonSeg dataset.

**Effect of the components of PPM.** In Table 6, comparing Exp. 3, Exp. 6, we observe that dynamic layer selection yields a clear improvement (+2.07% gIoU and +5.02% cIoU ), highlighting the benefit of leveraging information across multiple layers. Moreover, the decomposition indicates that using the similarity map as a prompt ($\mathcal{M}_{prompt}$) contributes the most (*e.g.*, Exp. 1 vs. Exp. 2, and Exp. 4 vs. Exp. 5) within this paradigm.

*Table 6.* Ablation study on PPM.

| Exp. ID | SSC | $<$SEG$>_{prompt}$ | $\mathcal{M}_{prompt}$ | gIoU | cIoU |
|---|---|---|---|---|---|
| 1 | | ✓ | | 19.13 | 15.62 |
| 2 | | | ✓ | 60.30 | 53.86 |
| 3 | | ✓ | ✓ | 64.06 | 67.05 |
| 4 | ✓ | ✓ | | 31.76 | 34.08 |
| 5 | ✓ | | ✓ | 52.14 | 52.42 |
| 6 | ✓ | ✓ | ✓ | **66.13** | **72.07** |

**Effect of explicit constraints for similarity map $\mathcal{M}$.** In Table 7, Exp. 1, 2, and 5 optimize the similarity map against hard labels (binary masks), while Exp. 3, 4, and 6 optimize it against soft labels (continuous masks). The results demonstrate that converting the ground-truth mask into a Gaussian heatmap smooths the object boundaries, leading to performance gains of +2.21% gIoU and +3.52% cIoU when comparing Exp. 5 and Exp. 6.

*Table 7.* Ablation study on loss constraints for similarity map $\mathcal{M}$.

| Exp. ID | BCE | Dice | Gaussian BCE | Gaussian Dice | gIoU | cIoU |
|---|---|---|---|---|---|---|
| 1 | ✓ | | | | 64.56 | 71.44 |
| 2 | | ✓ | | | 64.57 | 68.63 |
| 3 | | | ✓ | | 63.16 | 65.74 |
| 4 | | | | ✓ | 65.80 | **73.49** |
| 5 | ✓ | ✓ | | | 63.92 | 68.55 |
| 6 | | | ✓ | ✓ | **66.13** | 72.07 |

**Effect of policy reward components.** We use the negative loss of the similarity map relative to the ground-truth mask as the reward. During training, it is treated as a scalar and detached from the computation graph. In Table 8, comparing Exp. 1 and Exp. 2, we observe that using $r_{bce}$ as the reward signal outperforms using $r_{dice}$, yielding a gain of +2.03% gIoU and +1.20% cIoU.

*Table 8.* Ablation study on reward.

| Exp. ID | $r_{bce}$ | $r_{dice}$ | gIoU | cIoU |
|---|---|---|---|---|
| 1 | ✓ | | 65.37 | 69.53 |
| 2 | | ✓ | 63.34 | 68.33 |
| 3 | ✓ | ✓ | **66.13** | **72.07** |

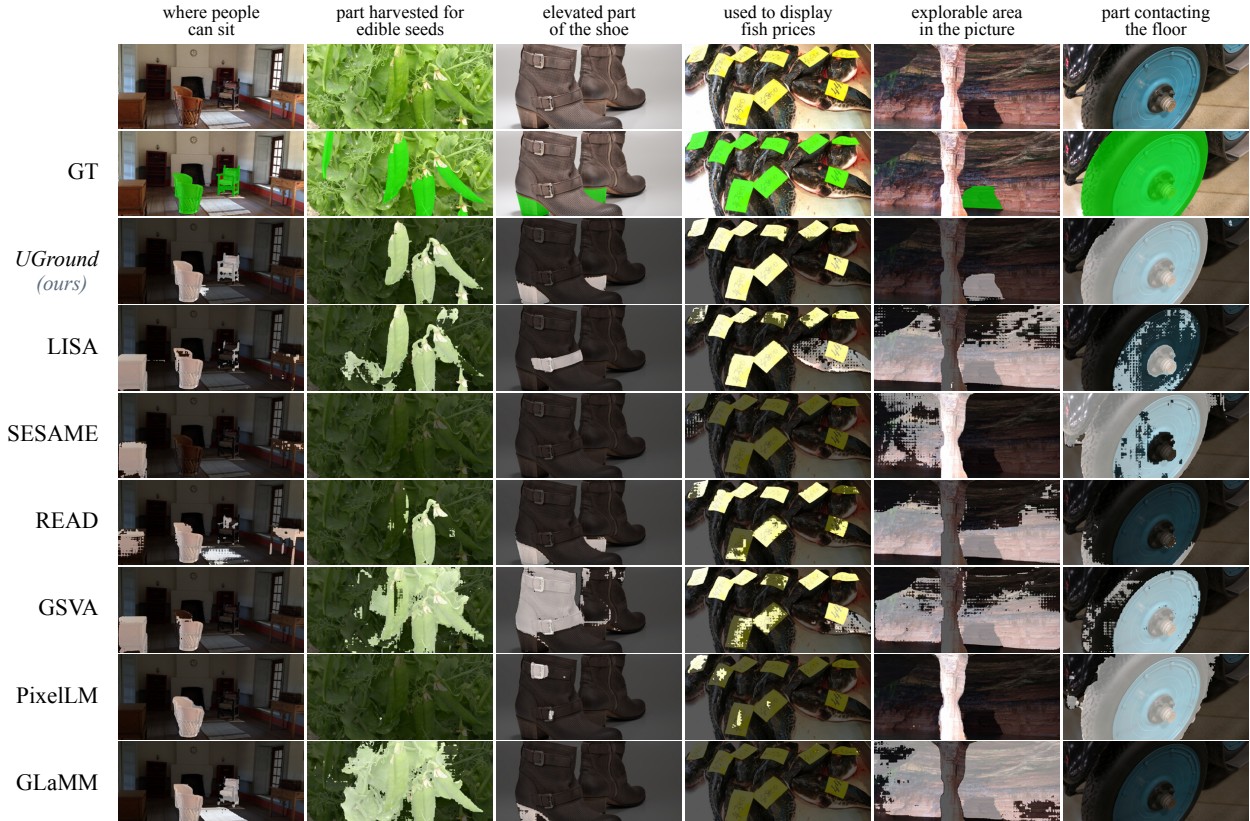

*Figure 4.* Visual comparison of UGround (Ours) with prior works on the ReasonSeg *test* set.

## 5.4. Qualitative Results

In Figure 4, UGround generates more accurate and semantically consistent segmentation masks than existing methods, including LISA (Lai et al., 2024), SESAME (Wu et al., 2024), READ (Qian et al., 2025), GSVA (Xia et al., 2024), PixelLM (Ren et al., 2024), and GLaMM (Rasheed et al., 2024). In challenging cases that require functional or commonsense grounding, such as "where people can sit" and "part harvested for edible seeds," UGround accurately localizes the target regions while avoiding irrelevant background regions. Moreover, UGround exhibits stronger fine-grained segmentation capability for small or part-level targets, such as "the elevated part of the shoe" and "the region used to display fish prices." Compared with competing methods that often produce incomplete, noisy, or over-expanded masks, UGround yields cleaner boundaries and better alignment with the ground-truth masks.

## 6. Conclusion

In this work, we first visualize all unrolled Transformer layers of LMMs, represented by LLaVA, and analyze the influence of intermediate layers on downstream task decoding. Building on this analysis, we present UGround, which features a dynamic layer selection and prompting mechanism—Policy-Prompted Masking (PPM). PPM enables intermediate layers to connect to the SAM decoder in a skip-connection-like manner and adjust its connectivity in a dropout-like fashion under the "mask as prompt" paradigm, together contributing to unified visual grounding. Our future work aims to further investigate how the <SEG> token interacts with the integrated SAM module for alignment.

**Acknowledgments.** The computations in this research were performed using the CFFF platform of Fudan University.

## Impact Statement

This paper presents UGround to advance a unified paradigm for visual grounding. By dynamically selecting intermediate layers in unrolled transformers and using "mask as prompt" to provide explicit spatial cues to a segmentation model (*e.g.*, SAM), UGround supports a broad range of settings within one system, spanning traditional referring expression segmentation to reasoning segmentation, single-target to multi-target queries, and rejection under false-premise (empty-target). In terms of societal impact, visual grounding can benefit human–AI interaction for image understanding, data annotation, and content editing. However, they may also be misused for privacy-invasive or surveillance applications (*e.g.*, automatically localizing and extracting sensitive regions in personal imagery), or to facilitate the creation and dissemination of misleading content (*e.g.*, precise cutouts and compositing of specific subjects). To mitigate these risks, we adhere to established community standards for responsible deployment. In particular, we encourage auditability in real-world applications and transparency in model design to foster trust.

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

# A. Notations

The notations used throughout this paper are summarized in Table 9 for clarity.

*Table 9.* Key notations used in this paper.

| Notation | Description |
|---|---|
| $\mathcal{G}_\theta$ | LMMs-based segmentation model, including a multi-modal LLM $\mathcal{G}_\mathcal{T}$ and a visual backbone model $\mathcal{G}_\mathcal{V}$, $\mathcal{G}_\theta = \mathcal{G}_\mathcal{V} \circ \mathcal{G}_\mathcal{T}$ |
| $\mathcal{H}^{(\ell)}$ | the hidden states at the $\ell$-th transformer layer |
| $\boldsymbol{h}_{t^*}^{(\ell)}$ | the hidden state of `<SEG>` token at layer $\ell$-th (position $t^*$) |
| $\hat{\mathbf{M}}$ | the predicted binary segmentation mask |
| $M$ | the ground-truth mask |
| $M_\sigma$ | the softened label obtained by applying a Gaussian kernel to $M$ |
| $\varphi(\cdot)$ | a multilayer perceptron (MLP) projection layer |
| $\mathcal{M}_{\text{prompt}}$ | mask as a prompt for SAM decoder $\mathcal{G}_\mathcal{V}^{dec}$ |
| `<SEG>`$_{\text{prompt}}$ | `<SEG>` token as prompt for SAM decoder $\mathcal{G}_\mathcal{V}^{dec}$ |
| $\mathcal{M}_{\text{IoU}}$ | the overlap between the similarity map and the ground-truth mask |
| $\mathcal{S}^{(\ell)}$ | the similarity scores at the layer $\ell$-th, |
| $\mathcal{M}$ | the soft logit mask or similarity map $\mathcal{M} \in [0,1]^{H \times W}$ |
| $\pi_\theta(\cdot \mid \mathcal{H}_{t^*})$ | the policy distribution conditioned on $\mathcal{H}_{t^*}$ |
| $G_\sigma(x,y)$ | a 2D Gaussian kernel with $\sigma$, $G_\sigma(x,y) \propto \exp(-\frac{x^2+y^2}{2\sigma^2})$ |

# B. Analysis

**Complexity Analysis.** To justify the complexity of RL-based layer selection (RL-gating), we compare it against a simpler non-RL alternative—soft-gating, *i.e.*, a learned soft-attention mechanism that produces a weighted average over layer outputs. We report results on ReasonSeg val set. In Table 10, RL-gating consistently outperforms soft-gating, with gains of +1.47% in gIoU and +1.34% in cIoU.

*Table 10.* RL-gating vs. soft-gating.

| Exp. ID | Method | gIoU | cIoU |
|---|---|---|---|
| 1 | soft-gating | 64.66 | 70.73 |
| 2 | RL-gating | 66.13 | 72.07 |

**Runtime Analysis.** To assess the computational costs of UGround, we report training and inference latency on a single NVIDIA A100-SXM4-40GB GPU in Table 11 and Table 12. We train on the ReasonSeg train split over 400 samples for 1 epoch (200 steps), with a batch size of 2. Training Latency (s) refers to the average time per iteration during the 200-step training process (including forward, backward, and optimizer step), and Memory Usage refers to the MaxMemAllocated observed within the 200 steps. We also report the model's Trainable Percentage (Trainable Parameters / Total Parameters). For inference, we evaluate on the ReasonSeg test split over 799 samples, with a batch size of 1. Speed (FPS) refers to the average frames per second computed over the 799 samples. In Table 11, UGround exhibits a higher training latency (2.97s) than LISA (Lai et al., 2024)(1.26s) due to the explicit optimization of the similarity map during the backward pass. In Table 12, UGround achieves a comparable inference runtime to LISA (Lai et al., 2024) and SESAME (Wu et al., 2024).

*Table 11.* Comparing the training cost of our UGround to state-of-the-art methods.

| Model | Training Latency (s) | Memory Usage (GB) | Trainable (%) | Trainable Params | Total Params |
|---|---|---|---|---|---|
| SESAME (Wu et al., 2024) | 1.11 | 23.15 | 3.73% | 288.25M | 7.73B |
| PixelLM (Ren et al., 2024) | 1.22 | 23.02 | 5.25% | 375.72M | 7.16B |
| LISA (Lai et al., 2024) | 1.26 | 23.68 | 3.74% | 288.25M | 7.71B |
| GSVA (Xia et al., 2024) | 2.13 | 25.73 | 3.73% | 288.26M | 7.73B |
| UGround | 2.97 | 28.16 | 3.72% | 288.40M | 7.75B |

*Table 12.* Comparing the runtime speed of our UGround to state-of-the-art methods.

| Model | GSVA (Xia et al., 2024) | SESAME (Wu et al., 2024) | LISA (Lai et al., 2024) | PixelLM (Ren et al., 2024) | UGround |
|---|---|---|---|---|---|
| Speed (FPS) | 3.98 | 4.64 | 4.68 | 9.24 | 4.12 |

**Generalization Analysis.** To demonstrate broader generalization of UGround, we report results on MUSE benchmark. MUSE aims for multi-target reasoning segmentation (Ren et al., 2024). In Table 13, UGround-7B consistently outperforms PixelLM-7B across all splits of the MUSE benchmark. It improves the Val overall cIoU by +5.7% and boosts the challenging "few-targets" cIoU by +10.5%. On the Test overall set, UGround also achieves a +4.1% cIoU gain, yielding a clear performance margin over PixelLM-7B. Also, we report results on augmented FP-Refcoco(+/g) under false premises. In Table 14, UGround outperforms READ across all FP-Refcoco(+/g) splits, with gains of +1.59% to +2.99% in "see" accuracy and +1.30% to +2.43% in segment scores (cIoU).

*Table 13.* Comparison on MUSE benchmark. Numbers are cited from PixelLM (Ren et al., 2024).

| Method | w/o SAM | Val overall | | Test few targets | | Test many targets | | Test overall | |
|---|---|---|---|---|---|---|---|---|---|
| | | gIoU | cIoU | gIoU | cIoU | gIoU | cIoU | gIoU | cIoU |
| SEEM (Zou et al., 2023b) | ✓ | 13.6 | 16.2 | 23.6 | 24.9 | 8.5 | 13.2 | 11.7 | 15.7 |
| LISA-7B (Lai et al., 2024) | ✗ | 18.8 | 29.0 | 24.7 | 36.5 | 9.6 | 24.5 | 12.8 | 27.1 |
| LISA-7B$_{rec}$ (Lai et al., 2024) | ✗ | 24.5 | 31.1 | 30.0 | 30.9 | 12.4 | 23.2 | 16.2 | 24.8 |
| LISA-7B$_{aug}$ (Lai et al., 2024) | ✗ | 42.0 | 46.1 | 43.5 | 52.0 | 37.7 | 42.3 | 38.9 | 44.4 |
| PixelLM-7B (Ren et al., 2024) | ✓ | 42.6 | 50.7 | 44.6 | 59.2 | 37.7 | 42.8 | 39.2 | 46.3 |
| UGround-7B | ✓ | **52.4** | **56.4** | **46.2** | **69.7** | **38.8** | **46.9** | **40.1** | **50.4** |

*Table 14.* Comparisons of the state-of-the-art "see" and "segment" results on augmented FP-Refcoco(+/g) *val* set. "FP"(False Premise) denotes a query for an object absent from the image.

| Method | FP-RefCOCO | | FP-RefCOCO+ | | FP-RefCOCOg | |
|---|---|---|---|---|---|---|
| | See | Segment | See | Segment | See | Segment |
| LISA-7B (Lai et al., 2024) | 51.36 | 44.00 | 51.32 | 39.62 | 51.25 | 39.64 |
| Cascading (Wu et al., 2024) | 75.59 | 55.18 | 75.03 | 48.64 | 76.07 | 49.98 |
| SESAME (Wu et al., 2024) | 79.84 | 57.93 | 80.00 | 50.81 | 81.78 | 53.79 |
| READ (Qian et al., 2025) | 82.87 | 61.50 | 83.51 | 54.54 | 84.67 | 56.12 |
| UGround | **85.86** | **62.80** | **85.10** | **56.03** | **86.86** | **58.55** |

# C. Discussions

## C.1. Justification for Reinforcement Learning (RL)

Reinforcement Learning (RL) is introduced to address the hard-boundary problem in layer selection, enabling the model to explicitly choose a layer (*e.g.*, layer 26) in a soft, differentiable manner during training. Because hard-boundary layer selection is non-differentiable, using RL is necessary.

An intuitive approach is to train a classifier for dynamic layer selection using cross-entropy loss, followed by soft-gating via a weighted average over all layer outputs. We call this simpler alternative CE + soft-gating. Here, the layer-selection label is assigned based on the overlap (IoU) between each layer's similarity map (layers 0–31) and the ground-truth mask. The classifier is trained, and its output scores serve as soft gates to weight and fuse the transformer layers (0–31) into a unified representation, which is then fed into SAM for mask decoding. However, CE + soft-gating suffers from the following issues:

1) CE + soft-gating is numerically suboptimal. Consider a scenario after training for 1–2 epochs where the classifier predicts the following scores for layers 0–31: $([0.01, 0.004, \ldots, 0.7, 0.1, 0.1])$. Since 0.7 is the maximum score, the 29-th layer is identified as the best layer. However, if we use the full score vector $([0.01, 0.004, \ldots, 0.7, 0.1, 0.1])$ as soft-gating weights to fuse the embeddings of layers 0–31, the best embedding from layer 29 will inevitably be mixed with features from all other layers (0–28 and 30–31), making it no longer optimal. Soft-gating inherently prevents the remaining layers from receiving zero weight. In fact, as shown in Fig. 5 ( and also discussed in (Zhang et al., 2025)), transformer layers are activated only at a few specific depths. Assigning non-zero weights to inactive layers dilutes the contribution of these key activated layers. Therefore, a hard boundary is required—that is, explicitly selecting a single layer (*e.g.*, assigning 100% weight to layer 29) rather than distributing fractional weights (*e.g.*, 60% on layer 29). This is inherently a discrete decision problem, for which reinforcement learning (RL) provides a principled and trainable solution.

2) The training objective of the layer-selection classifier (*i.e.*, CE loss) does not align with the ultimate goal of optimizing the gates through backpropagation from the downstream SAM mask decoding (*i.e.*, use soft-gating without an explicit loss on the gates). The layer-selection classifier explicitly trains the gate to predict which layer is the most important, whereas using soft weights to fuse all layers aims to optimize the averaged representation, without explicitly enforcing the gate to select the optimal layer. Besides, soft-gating presumably ignores inter-layer dependencies in the transformer. By weighting and summing all layers, it actually performs a soft ensemble, disrupting the original layer-by-layer structure of the transformer.

3) Computational cost: CE + soft gating vs. RL-gating. Given $\boldsymbol{h}_{seg} \in \mathbb{R}^{4096}$ and $\boldsymbol{h}_{img} \in \mathbb{R}^{576 \times 4096}$, the similarity map is computed as: $\boldsymbol{h}_{seg} \times \boldsymbol{h}_{img}^T = (1 \times 4096) \times (576 \times 4096)^T = 576$, followed by reshaping into $24 \times 24$, and then interpolating to $256 \times 256$ to obtain $\mathcal{M}$. If using the CE + soft-gating approach, one must compute the similarity map for all 32 layers of LLaVA, resulting in $32 \times 256 \times 256$ maps, and then compute the overlap (IoU) with the ground-truth mask as the supervision label. In contrast, with RL, we compute $\boldsymbol{h}_{seg} \times \boldsymbol{w}^T = (1 \times 4096) \times (32 \times 4096)^T = 32$, yielding a 32-dimensional similarity logits vector, followed by REINFORCE to select the layer index, *e.g.*, $(\ell^* = 28)$. $\boldsymbol{w}^T \in \mathbb{R}^{32 \times 4096}$ denotes the layer-gating weights, which are the only learnable parameters in UGround. The reward $r$ only requires computing the similarity map for a single layer.

### C.2. Costs of Mask as Prompt

Compared to the `<SEG>` token, Mask as Prompt (MasP) is more effective, as it provides explicit spatial cues via its activation regions. Here, we analyze the computational costs associated with Mask as Prompt. We use SAM (Kirillov et al., 2023) for mask decoding. The original paper describes its prompt encoder as follows:

*"Prompt encoder. We consider two sets of prompts: sparse (points, boxes, text) and dense (masks). We represent points and boxes by positional encodings summed with learned embeddings for each prompt type, and free-form text with an off-the-shelf text encoder from CLIP (Radford et al., 2021). Dense prompts (e.g., masks) are embedded using convolutions and summed element-wise with the image embedding."*

*Table 15.* Mask prompt dimension changes through convolutional layers in SAM.

| Layer | Output shape |
|---|---|
| Input mask: $256 \times 256$ | (B, 1, 256, 256) |
| Conv2d($1 \rightarrow 4$, stride=2) | (B, 4, 128, 128) |
| Conv2d($4 \rightarrow 16$, stride=2) | (B, 16, 64, 64) |
| Conv2d($16 \rightarrow 256$, stride=1) | (B, 256, 64, 64) |

That is, SAM accepts a mask prompt of size $256 \times 256$, which is transformed by the convolutional layers in Table 15. Thus, the final mask embedding has dimensions [B, 256, 64, 64]. In SAM's implementation, even when the mask-as-prompt input is not provided, *i.e.*, when only the text `<SEG>` prompt is used (`mask=None`), SAM still generates a set of learnable embeddings of the same shape as a placeholder, serving as the mask-embedding representation. Therefore, using Mask as Prompt does not introduce any additional parameter overhead.

### C.3. Analogous to Monte Carlo Dropout

Analogous to Monte Carlo Dropout. Consider the query: "Can you segment the apple, orange, and banana in the picture ?" UGround responds: "Yes, apple, orange, and banana are `<SEG>`, `<SEG>`, `<SEG>`." That is, there are three objects to segment in a single image. During one forward pass, our SSC module (RL) selects a layer from 0–31 for each of `<SEG>`$_{apple}$, `<SEG>`$_{orange}$, and `<SEG>`$_{banana}$ as input to SAM. Suppose: in the (1)-th forward pass, layers (15, 28, 19) are selected, that is, `<SEG>`$^{(15)}_{apple} \rightarrow$ SAM, `<SEG>`$^{(28)}_{orange} \rightarrow$ SAM, `<SEG>`$^{(19)}_{banana} \rightarrow$ SAM; in the ($\mathcal{T}$-1)-th forward pass, the selections are `<SEG>`$^{(16)}_{apple} \rightarrow$ SAM, `<SEG>`$^{(20)}_{orange} \rightarrow$ SAM, `<SEG>`$^{(14)}_{banana} \rightarrow$ SAM; in the $\mathcal{T}$-th forward pass, the selections are `<SEG>`$^{(13)}_{apple} \rightarrow$ SAM, `<SEG>`$^{(17)}_{orange} \rightarrow$ SAM, `<SEG>`$^{(26)}_{banana} \rightarrow$ SAM, then during the $\mathcal{T}$ forward passes, `<SEG>`$_{apple}$ from layers 15, 16, and 13 are independently connected to SAM, and the SSC module dynamically alters the model's internal connectivity. This can be analogized to the dropout mechanism in neural networks: All layers are virtually connected to SAM, but only one path is activated per forward pass. Since the SSC module performs stochastic sampling, this can be interpreted as a Monte Carlo uncertainty estimation (Gal & Ghahramani, 2016). Unlike soft-gating, the SSC module enables each layer to connect independently to SAM, resulting in 32 independent subnetworks in an ensemble-style

design. As only one layer is activated at a time, the computational overhead is comparable to selecting a fixed last layer, thus avoiding expensive computational costs.

## D. Qualitative Results

**Layer-wise visualization of the Transformer across layers 1-40.** Fig. 5–7 present a qualitative analysis of similarity map activations across layers 1–40, illustrating their alignment with the ground-truth mask. Each figure contrasts heatmaps generated by prior works using a fixed last hidden layer (a) with those from our proposed Policy-Prompted Masking method (b). Unlike the baseline approach (a), our method introduces stochastic layer selection: in each forward pass, a layer $\ell^*$ is randomly chosen, and multiple $\mathcal{T}$ stochastic samples result in an approximate Bayesian uncertainty estimation, akin to Monte Carlo Dropout. This sampling mechanism explores diverse pathways, alleviates over-reliance on any single trajectory, and virtually forms an ensemble of subnetworks, thereby enhancing robustness. As a result, the similarity maps in (b) show activations that are sharply concentrated on the target object in intermediate and later layers, yielding more accurate alignment with the ground-truth mask compared to the diffuse and noise-prone activations observed in (a).

**Visualization of UGround on benchmarks.** Fig. 8–9 show qualitative results on ReasonSeg test set. Fig. 9 demonstrates UGround's capability in semantic segmentation, *i.e.*, assigning pixel-level labels for a target semantic category (with background as 0). Fig. 10–11 show qualitative results on gRefCOCO (Liu et al., 2023) test sets A and B, which demonstrate UGround's capability in instance segmentation, *i.e.*, predicting instance-level masks for individual object instances (with separate masks for objects of the same category). Fig. 12 shows qualitative results on FP-RefCOCO(+/g) val set, which demonstrates UGround's capability in handling false-premise queries, *i.e.*, queries that refer to objects not present in the given image (empty target). Notably, UGround is capable of correcting false premises, *i.e.*, directly refusing queries when the target object is absent and optionally suggesting look-alike objects (Wu et al., 2024) as shown in Fig. 12.

## E. Implementation Details

**The UGround code repository.** Currently, UGround supports 8 dataset types, namely, A: sem_seg, B: refer_seg, C: neg_refer_seg, D: correct_refer_seg, E: vqa, F: reason_seg, G: reason_seg_plus, and H: multi_reason_seg. In particular,

A: sem_seg: ade20k || cocostuff || pascal_part || paco_lvis || mapillary

B: refer_seg: refclef || refcoco || refcoco+ || refcocog || refzom || grefcoco

C: neg_refer_seg: R-Refcoco || R-Refcoco+ || R-Refcocog

D: correct_refer_seg: fprefcoco || fprefcoco+ || fprefcocog

E: vqa: llava_instruct_150k

F: reason_seg: ReasonSeg|train

G: reason_seg_plus(LISA++): instance_seg || cot || conversations || caption

H: multi_reason_seg(muse): MultiReasonSeg|train

Our UGround is trained in a mixed fashion on these 8 types of "in-the-wild" datasets as needed. As shown in Fig. 14, the UGround project is organized with a model directory for core implementations, a scripts directory containing training and testing scripts, and a configs directory for configuration files. The main model architecture is defined in UGround.py within the UGroundForCausalLM class, which extends LlavaLlamaForCausalLM and integrates a Segment Anything Model (SAM) for mask generation. A central component of this architecture is the PolicyPromptedMasking class, detailed in PPM.py, which is instantiated and used by the UGroundForCausalLM class to dynamically select hidden state features from different network layers. The selection policy is trained using the REINFORCE algorithm with either an EMA or a critic-based baseline. All codes and models are publicly available at https://github.com/rui-qian/UGround.

**The UGround dashboard.** As shown in Fig. 13, we developed a web-based dashboard to showcase the capabilities of UGround. The system accepts multimodal inputs, specifically a user-uploaded image paired with a natural language instruction. In response, the interface generates two outputs: (1) a segmentation mask that highlights the image region corresponding to the instruction, and (2) a textual explanation that answers the user's query.

# F. Ethics and Societal Impact

**Use of Large Language Models.** During the preparation of this manuscript, we used Large Language Models (LLMs) primarily to aid in writing and polishing the text. Specifically, LLMs were employed for grammar correction, spelling improvements, and sentence restructuring to enhance readability and clarity. In terms of technical content, LLMs assisted only in routine tasks such as selecting tensor indices and writing dataset DataLoader code; all algorithmic ideas, methodology, and experimental designs were developed entirely by the authors. All model-generated text was reviewed and edited by the authors, who take full responsibility for the final content of this paper.

# G. Limitations and Future Work

We observe that UGround generally understands which object is being referred to, which means that the semantic grounding is correct (*i.e.*, the language responses are accurate), yet the predicted masks often fail to capture the full spatial extent of the object. The masks are frequently fragmented, perforated with holes, spatially offset, or spilled into irrelevant regions; in some cases, they even collapse entirely. This indicates that the issue does not stem from semantic misinterpretation, but from imperfect cross-modal geometric alignment.

Although the <SEG> token is intended to represent the "target object," it is fundamentally a language-space embedding whose feature distribution is not inherently aligned with the spatial structure of visual tokens. In other words, <SEG> token $\neq$ spatially aligned embedding, embedding space $\neq$ image space. For example, a distant patch of blue sky and a blue notebook on a desk may appear close in embedding space due to similar color or texture, even though they are far apart in the image. Thus, the <SEG> token captures semantic concepts rather than the spatial statistics of an object's distribution. While using the mask as a prompt does introduce spatial cues, the internal decoding mechanism of SAM remains largely unexplored, making it unclear how these cues propagate into its actual segmentation behavior.

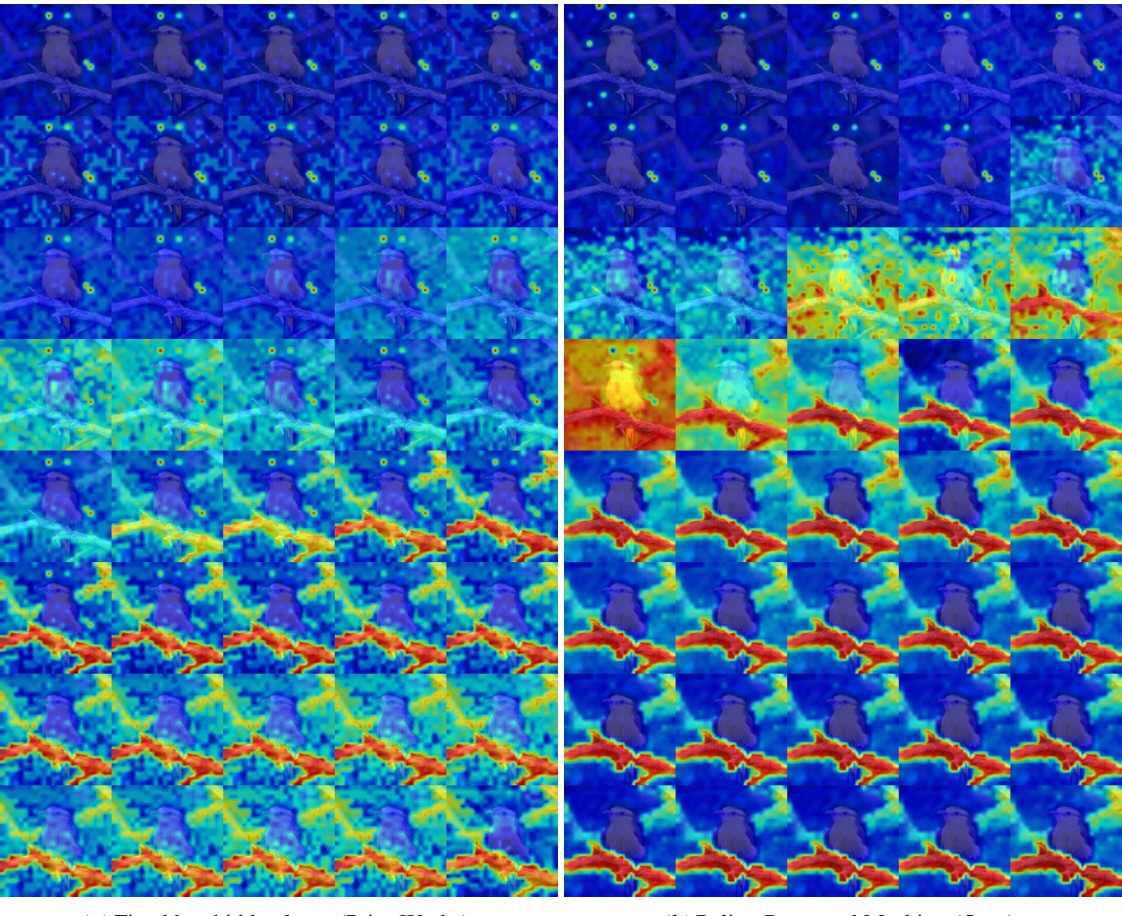

(a) Fixed last hidden layer (Prior Works).   (b) Policy-Prompted Masking (Ours).

*Figure 5.* Similarity map across layers 1-40.

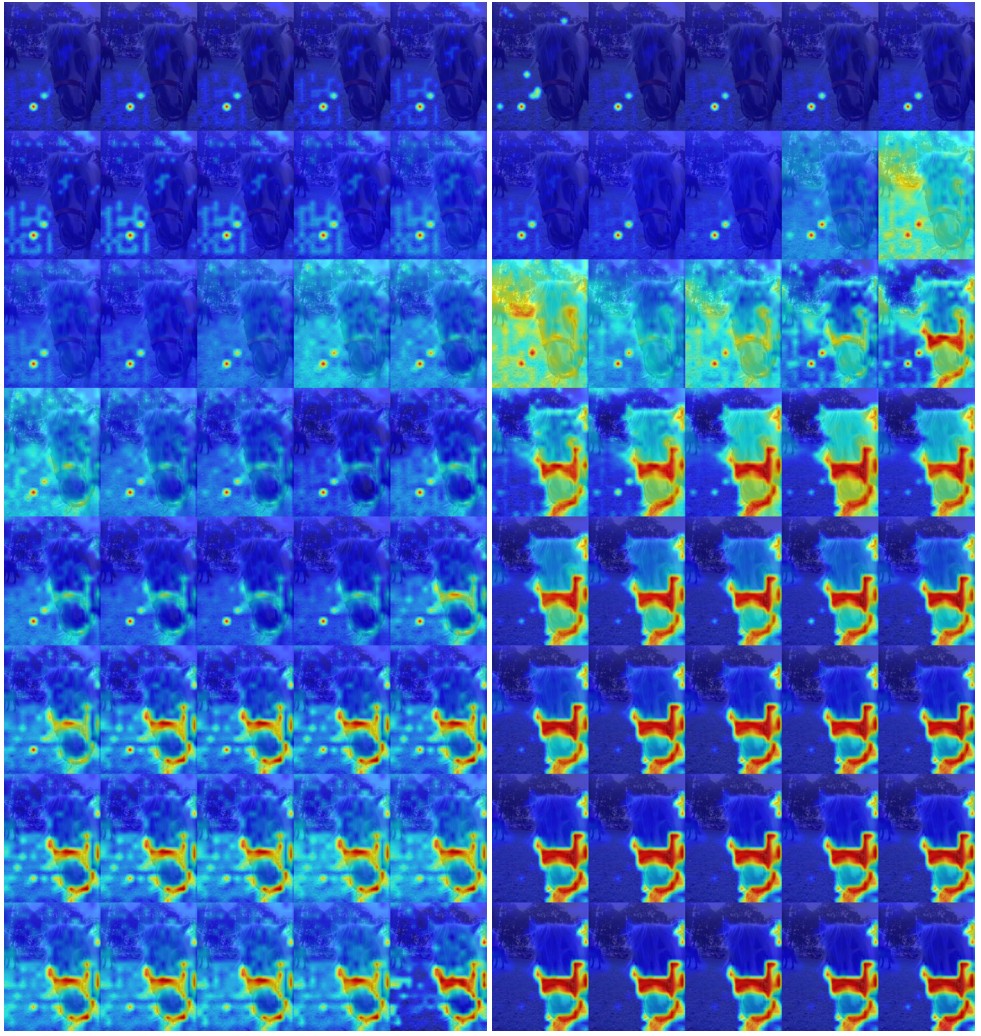

*(a)* Fixed last layer (Prior Works).    *(b)* Policy-Prompted Masking (Ours).

*Figure 6.* Similarity map across layers 1-40.

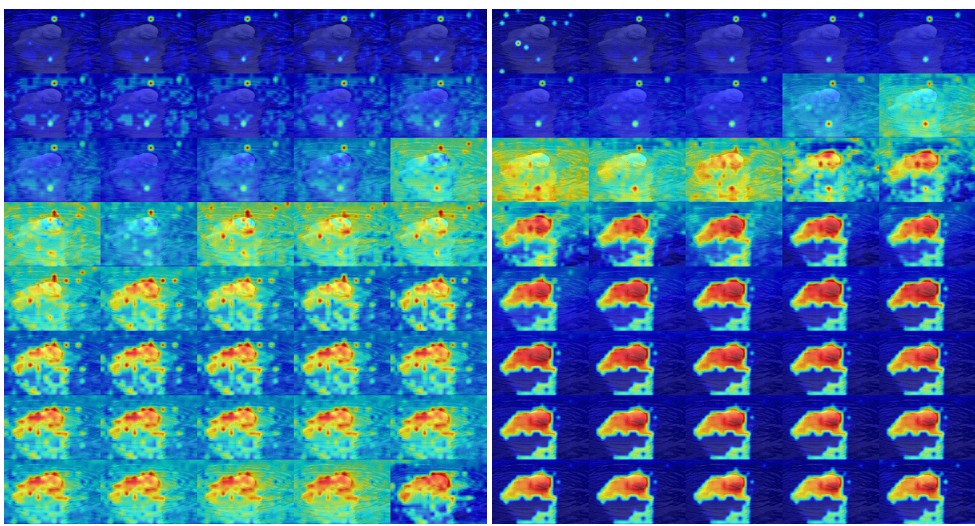

*(a)* Fixed last hidden layer (Prior Works).    *(b)* Policy-Prompted Masking (Ours).

*Figure 7.* Similarity map across layers 1-40.

**User:** When exploring the beach and handling sea creatures, we must be cautious to avoid injuries. What in the picture can *cause injuries if touched* without proper care?

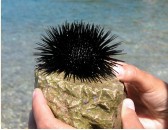
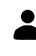

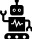

**Reasoning:** Sure, it is [SEG]. In the image, a person is holding a sea creature with a large, spiky head, which appears to be a sea urchin. *Sea urchins* are known for their *sharp spines*, which can cause injuries if not handled with care. When exploring the beach and coming into contact with these sea creatures…

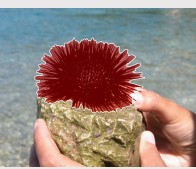

**User:** What are the *bird that has caught prey* in this image?

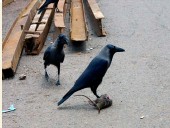
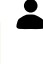

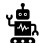

**Reasoning:** It is [SEG]. The bird in the picture is a black bird, possibly *a crow or a raven*, that is standing on the ground with its catch, which appears to be *a small animal or a mouse*. The other two birds are perched on a wooden structure…

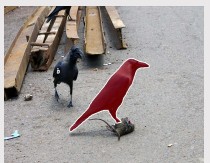

**User:** Navigating on the sea can often be dangerous due to various hazards. What in the picture could *pose a threat to humans*?

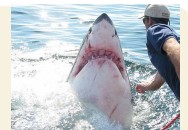
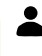

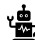

**Reasoning:** Sure, it is [SEG]. In the image, a man is standing near *a large, open mouth shark* in the water. The presence of a shark, a powerful predator, can *be a potential threat to humans*, as they are known to be aggressive and can inflict severe injuries or even fatalities when they attack…

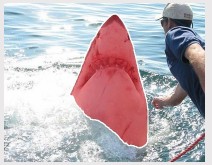

**User:** When an aircraft is ready to depart from the airport, where does it usually *begin its takeoff*?

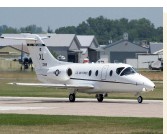
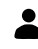

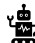

**Reasoning:** It is [SEG]. The designated area for an aircraft to *begin its takeoff is the runway*, which is a long, flat, and usually paved surface at the airport. In the image, the white and blue light aircraft is on the runway, preparing for takeoff…

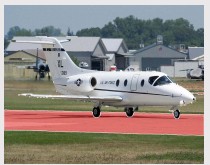

**User:** In many countries, it is illegal to park a car in certain areas without proper permission. What part of the picture can help *identify the ownership and registration of the car* to ensure it is legally parked?

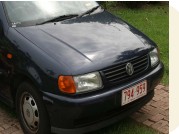
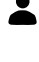

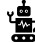

**Reasoning:** Sure, it is [SEG]. In the image, the car is parked on the side of the road with its *license plate* visible. The license plate is essential for *identification and registration purposes*. It is used to determine the legal ownership of the car…

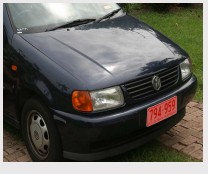

*Figure 8.* Visualization of UGround-7B on the ReasonSeg test set, demonstrating UGround's capability in semantic segmentation, *i.e.*, assigning pixel-level labels for a target semantic category (with background as 0). Zoom in for best view.

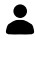

**User:** If you were working in a bakery, where in the picture could you *place bread and pastry products to cool them down* after they come out of the oven? 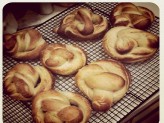

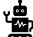

**Reasoning:** Sure, it is [SEG]. In the image, there is a cooling rack with a tray on it, which is used to *cool down the bread and pastry products after they exit the oven*. The cooling rack is an essential tool in a bakery, as it helps to evenly distribute heat and ensure that the baked goods don't become too greasy or lose their texture… 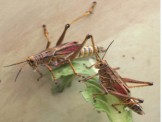

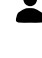

**User:** The application of pesticides to crops might be necessary to prevent infections in agricultural administration. What part in the picture is *at risk* during this process? 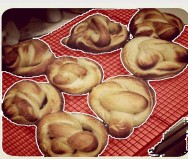

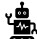

**Reasoning:** It is [SEG]. In the image, there are *two large grasshoppers on a plant*, which is a part of the crop. When pesticides are applied to the crops, it is crucial to be cautious with the presence of these grasshoppers, as they can damage the crop and reduce the yield. The pesticides should be applied in a way that targets the grasshoppers specifically, while minimizing the risk to other beneficial insects and the environment. 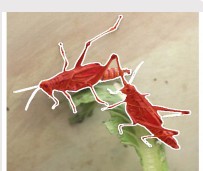

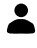

**User:** What is the primary musical instrument being played by the individual in the picture during a football game as the marching band seeks *to energize the team and engage the crowd*? 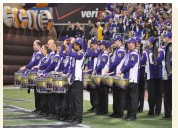

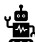

**Reasoning:** Sure, it is [SEG] . In the image, the man is playing a drums, which is a primary musical instrument. *The marching band is using this instrument, along with other drums, to create a lively and engaging performance* during the football game. The band's performance aims to energize the team and engage the crowd, as they perform in front of a large audience. 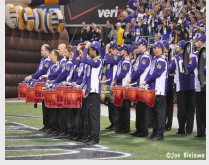

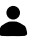

**User:** During a birthday party, there is often a table set up with various presents for the birthday boy or girl. What in the picture could be *used to hold and present these gifts* ? 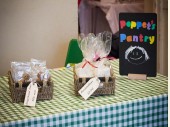

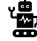

**Reasoning:** Sure, the segmentation result is [SEG] . In the image, there is a table with a green and white tablecloth, and on it, there are two baskets and a chalkboard. *The baskets can be used to hold and present the gifts for the birthday boy or girl*. The chalkboard can be used to write the birthday person's name, age, or other information related to the party. The tablecloth and the overall setting create a visually appealing and organized space for the birthday celebration. 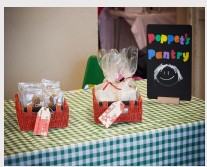

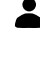

**User:** What might be the object present in the picture that is likely to *produce the pleasant scent of lavender*, which is commonly utilized in various products like soap and essential oils? 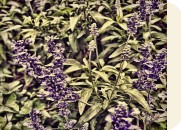

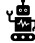

**Reasoning:** Sure, it is **[SEG]** . In the image, *there are a group of purple flowers, which are likely to be the source of the pleasant scent of lavender*. These flowers are often used in various products such as soap, essential oils, and other fragrant items due to their soothing and relaxing properties. The presence of the flowers in the picture suggests that they are in full bloom, releasing their sweet aroma into the air. 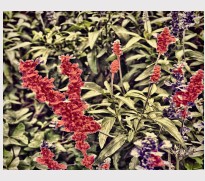

*Figure 9.* Visualization of UGround-13B on the ReasonSeg test set, demonstrating UGround's capability in semantic segmentation, *i.e.*, assigning pixel-level labels for a target semantic category (with background as 0). Zoom in for best view.

**User:** What are umbrella lady, white shirt guy in front of woman with umbrella cant see his head, and person with checkered clothing in this image? Please output segmentation mask.

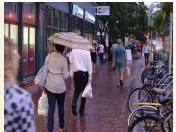 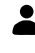

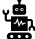

Umbrella lady is **[SEG]** , white shirt guy in front of woman with umbrella cant see his head is **[SEG]** and person with checkered clothing is **[SEG]** .

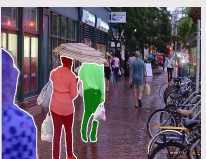

**User:** What are the cup of coffee (latte) with latte art, the glass of water and the teaspoon in this image ? Please output segmentation mask.

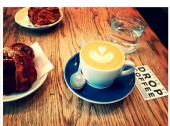 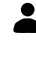

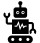

The segmentation result of the cup of coffee (latte) with latte art, the glass of water and the teaspoon are **[SEG]** , **[SEG]** and **[SEG]** .

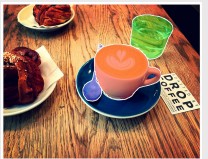

**User:** Can you segment the the man in red upon the zebra, the umpire and the man in red in this image?

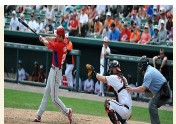 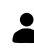

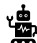

The segmentation result of the man in red upon the zebra, the umpire and the man in red are **[REJ]** , **[SEG]** and **[SEG]** .

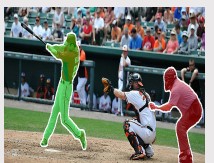

**User:** What are the wooden bench in the center, the light brown dog to the left of the bench, and the dog to the right of the bench in this image? Please output segmentation mask.

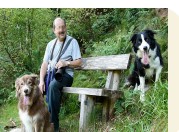 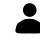

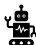

The segmentation result of the wooden bench in the center is **[SEG]** , the segmentation result of the light brown dog to the left of the bench is **[SEG]** and the segmentation result of the dog to the right of the bench is **[SEG]** .

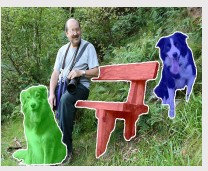

**User:** Please segment the blurry plate in the background and slice, the silver fork on the left side of the plate and sandwich on front plate in this image?

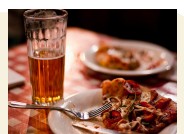 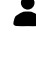

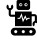

The segmentation result of blurry plate in the background and slice is **[SEG]** , the segmentation result of the silver fork on the left side of the plate is **[SEG]** and the segmentation result of sandwich on front plate is **[SEG]** .

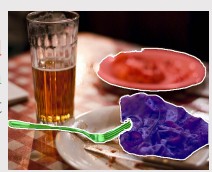

*Figure 10.* Visualization of UGround-7B on gRefCOCO (Liu et al., 2023) test sets A and B (I), demonstrating UGround's capability in instance segmentation, *i.e.*, predicting instance-level masks for individual object instances (with separate masks for objects of the same category). Zoom in for best view.

**User:** What are apple far left, orange far right and banana in this image? Please respond with segmentation mask. 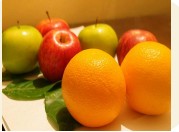 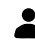

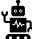 The segmentation result of apple far left is **[SEG]**, the segmentation result of orange far right is **[SEG]** and the segmentation result of banana is **[REJ]** . 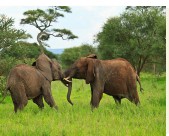 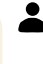

**User:** What are elephant on the right, left and car upper left in this image? Please output segmentation mask. 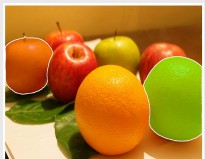 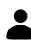

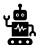 Elephant on the right is **[SEG]**, left is **[SEG]** and car upper left is **[REJ]** . 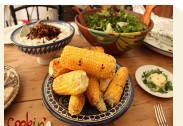

**User:** What are the bowl of food in the back right, the bowl in the front, and the bowl in the top left in this image? Please output the segmentation mask. 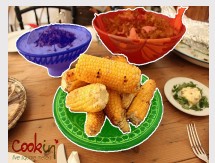 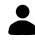

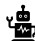 The segmentation result of the bowl of food in the back right is **[SEG]** , the segmentation result of the bowl in the front is **[SEG]** and the segmentation result of the bowl in the top left is **[SEG]** . 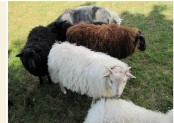

**User:** What are the brown sheep facing right, part of white goat lower right and center white sheep in this image? Please output segmentation mask. 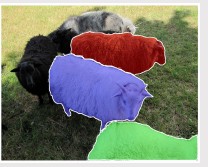 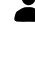

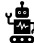 The segmentation result of the brown sheep facing right is **[SEG]**, the segmentation result of part of white goat lower right is **[SEG]** and the segmentation result of center white sheep is **[SEG]** . 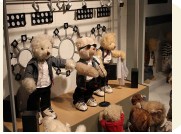

**User:** What are the three bears in this image: the one wearing sunglasses, the one to its right and the one to its left? Please output with segmentation mask. 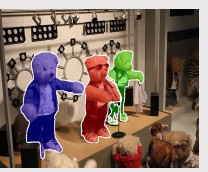 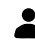

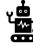 Sure, the segmentation result of the three bears in this image: the one wearing sunglasses, the one to its right and the one to its left are **[SEG]** , **[SEG]** and **[SEG]** .

*Figure 11.* Visualization of UGround-7B on gRefCOCO (Liu et al., 2023) test sets A and B (II), demonstrating UGround's capability in instance segmentation, *i.e.*, predicting instance-level masks for individual object instances (with separate masks for objects of the same category). Zoom in for best view.

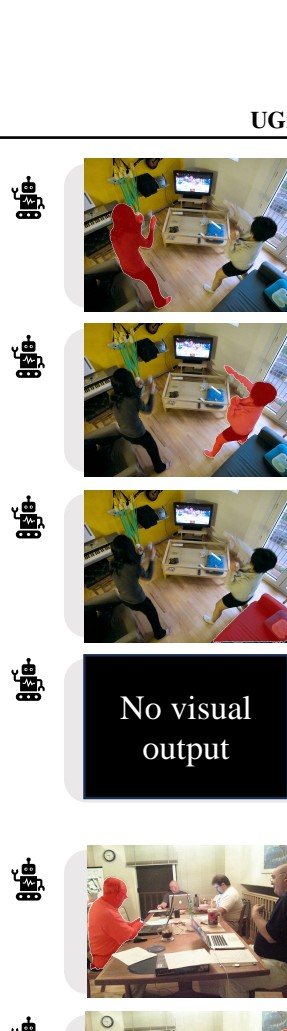

Yes, **person on the left** is here. Segmentation: [SEG].

Please inspect this image for **person on the left**. If it's there, share the segmentation map. If it's not, deny its existence and, if you see fit, list any objects that resemble person on the left ?

**Person on the right** is clearly in the picture. Segmentation: [SEG] .

Examine the image to verify if **person on the right** is present. If it is, provide the segmentation. If not, deny and, if it seems appropriate, note any objects that might be confused with person on the right.

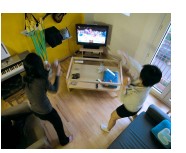

Yes, **blue couch** is here. Segmentation: [SEG] .

Examine the image to verify if **blue couch** is present. If it is, provide the segmentation. If not, deny and, if it seems appropriate, note any objects that might be confused with blue couch.

No visual output

I **can't** confirm the **person all in purple**, but the person all in black is certainly present.

Is there an occurrence of the **person all in purple** in this image? If yes, display the segmentation map. If no, reject and identify any resembling objects.

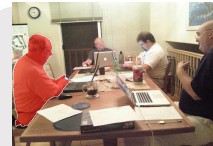

Yes, **left dude** is here. Segmentation: [SEG] .

Can you confirm the existence of **left dude** in this image? If it's there, provide a segmentation map. If not, deny and, if you see fit, list any objects resembling left dude.

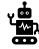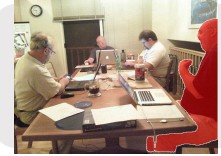

Yes, **guy on the far right** is here. Segmentation: [SEG] .

In this image, could you check for the occurrence of **guy on the far right**? If it's present, show the segmentation map. If not, decline and list similar-looking objects.

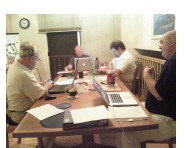

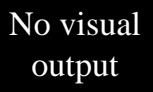

No visual output

**No left guy in black** here, but the image spotlights left guy in white.

Please inspect this image for **left guy in black**. If it's there, share the segmentation map. If it's not, deny its existence and, if you see fit, list any objects that resemble left guy in black.

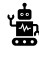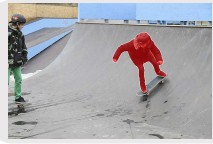

**Boy on skateboard** is clearly in the picture. Segmentation: [SEG] .

Assess the image for the occurrence of **boy on skateboard**. If it's detected, provide the segmentation map. If not, deny and, if relevant, point out any objects that may be confused with boy on skateboard.

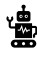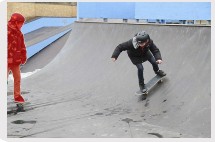

**Person on left edge** is clearly in the picture. Segmentation: [SEG] .

Can you confirm if **person on left edge** is present in this image? If it is, kindly provide the segmentation map. If it isn't, please state so and, if appropriate, mention any objects that might be mistaken for person on left edge.

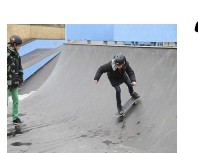

No visual output

The **kid with purple pants isn't here**, but check out kid with green pants.

Can you identify **kid with purple pants** in this image? If present, show the segmentation map. If absent, directly refuse and optionally suggest any look-alike objects.

*Figure 12.* Visualization of UGround-7B on the FP-RefCOCO(+/g) val set, demonstrating UGround's capability in handling false-premise queries, *i.e.*, queries that refer to objects not present in the given image (empty target). Zoom in for best view.

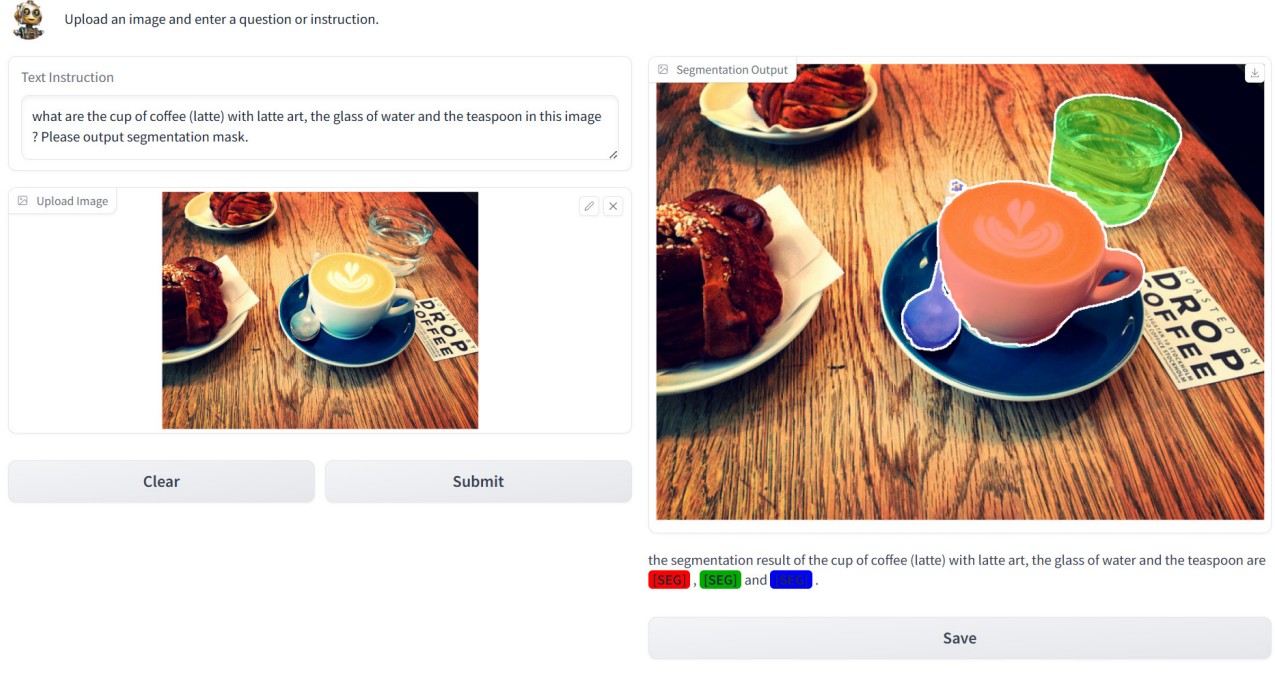

*Figure 13.* The UGround dashboard, demonstrating a unified interface that supports multiple visual grounding models, *i.e.*, LISA (Lai et al., 2024), SESAME (Wu et al., 2024), PixelLM (Ren et al., 2024), GSVA (Xia et al., 2024), and READ (Qian et al., 2025). This example illustrates the result of a single UGround inference.

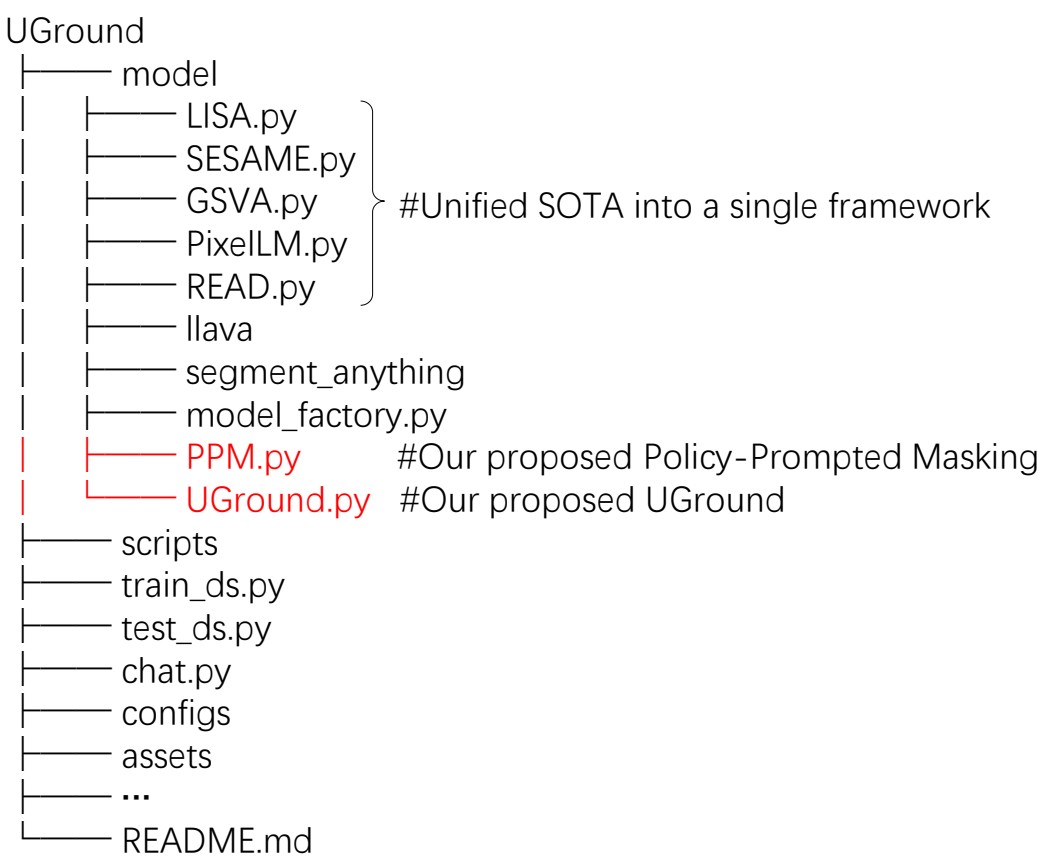

*Figure 14.* The UGround code repository. All codes and models are publicly available at https://github.com/rui-qian/UGround.

