# OpenReview forum: "UGround: Towards Unified Visual Grounding with Unrolled Transformers"
_ICML.cc/2026/Conference — ICML 2026 regular_

### Official Review · Reviewer_DbMw · 2026-03-11

**Soundness:** 3
**Presentation:** 3
**Significance:** 3
**Originality:** 3
**Overall Recommendation:** 4
**Confidence:** 5

**Summary:**

To address the issues of the traditional VG model lacking intermediate correction and explicit spatial cues, this paper proposes a unified framework based on unrolled transformer from the perspective of attributes, integrating refer expression segmentation and reasoning segmentation, and successfully resolves the problems of transitioning from single-object to multi-object and false premises.

**Compliance With Llm Reviewing Policy:**

Affirmed.

**Final Justification:**

My concerns are fully resolved.

**Key Questions For Authors:**

See weaknesses above.

**Limitations:**

No, more experiments should be conducted to discuss the possible limitations of the method.

**Strengths And Weaknesses:**

Strengths：The paper is generally well structured and easy to follow. The overall organization of the method and experiments is clear.

Weaknesses:
(1) In defining the unified framework, the author employs the term "attribute," which traditionally refers to concrete attributes such as color, shape, and material. However, its usage in this context is ambiguous and may lead to confusion.
(2) The author asserts that they have constructed a unified framework for the first time. Yet, single-objective, multi-objective, and no-objective configurations are already fundamental characteristics of generalized referring expression segmentation, a topic extensively explored in existing literature. Therefore, the claimed novelty of this unification requires further clarification.
(3) The core innovation of the paper lies in Policy-Prompted Masking, which comprises Stochastic Skip Connection (SSC) and Mask as Prompt (MasP). SSC dynamically selects transformer layers via reinforcement learning, while MasP guides SAM through similarity-based prompting. Although this design may enhance the model's focus on target regions, the overall contribution appears to be a modular combination of existing components.
(4) The experimental section lacks validation in terms of model parameters and inference speed. Additionally, the main text does not include visual experimental results. Although some visual experiments are provided in the appendix, they lack comparisons with other methods and fail to offer comparative validation of the key innovations.

---

> ### Author Rebuttal · Authors · 2026-03-28
>
> **AW1:** We will include the following for clarity: The term "attribute variation" (RES → RS/Multi-RS: explicit vs. implicit; RES → FP-RES: true vs. false; RES → gRES: single vs. multiple) is used because these characteristics can be readily extended to video, audio, point clouds, and other specialized tasks (e.g., detection). We generalize the term "attribute" beyond concrete visual properties (such as color, shape, and material) to denote inherent characteristics of task settings, such as query type, number of targets, and handling of false premises. The term "attribute variation" provides a fine-grained perspective across emerging visual grounding tasks.
>
> ---
> **AW2:**  We do construct a unified framework (Table 1, see also: [**[Links]**](https://github.com/anonymouslly/icml)), but from an attribute perspective (Lines 42-53; 126-133), which diverges from prior works that primarily focus on versatile capabilities, such as performing segmentation and captioning jointly (e.g., OMG-LLaVA, GLaMM), or grounding across both images and videos (e.g., HyperSeg, OMG-Seg). This difference lies in the granularity of task settings when defining "unified."  The attribute perspective is novel. As emerging visual grounding tasks continue to expand the capability boundaries of LLMs, it is crucial to clarify whether such extensions occur at the macro level (versatility) or the micro level (attribute), which in turn facilitates a more mature visual grounding taxonomy.
>
> Generalized referring expression segmentation (GRES) is ill-suited for complex (implicit) reasoning (Table 3; Line 340), as it is built upon explicit referring expressions derived from RES. In this context, "explicit → implicit" is what we term "attribute variation."
>
> Table 1: Attribute variation with respect to visual grounding. RES: Referring Expression Segmentation; RS: Reasoning Segmentation; FP: False Premise (reject queries with no target; **correct if target exists**); GRES: Generalized RES; Multi-RS: Multi-target RS.
>
> | Method | RES | RS | FP-RES | GRES | Multi-RS |
> |---|:---:|:---:|:---:|:---:|:---:|
> | LISA [CVPR'24] | ✓ | ✓ | ✗ | ✗ | ✗ |
> | SESAME [CVPR'24] | ✓ | ✓ | ✓ | ✗ | ✗ |
> | READ [CVPR'25] | ✓ | ✓ | ✓ | ✗ | ✗ |
> | GLaMM [CVPR'24] | ✓ | ✓ | ✗ | ✓ | ✗ |
> | GSVA [CVPR'24] | ✓ | ✓ | ✓ | ✓ | ✗ |
> | PixelLM [CVPR'24] | ✓ | ✓ | ✗ | ✓ | ✓ |
> | UGround (Ours) | ✓ | ✓ | ✓ | ✓ | ✓ |
>
> ---
> **AW3:** (1) Prior works (e.g., LISA, PixelLM, GSVA, GLaMM) rely on the fixed last hidden layer, while little prior work has explored intermediate layers in visual grounding. We employ RL-based dynamic layer selection (SSC) to bridge this gap. SSC introduces effects analogous to skip connections and dropout in computer vision, while remaining distinct from them in principle.
>
> (2) To our knowledge, little prior work uses similarity maps (a.k.a. soft logit masks) to directly prompt SAM. Such similarity maps are differentiable, enabling gradients from the SAM branch to backpropagate to the LLM. In contrast, existing works (e.g., OMG-LLaVA) treat masks as data features (analogous to xyz coordinates ) to fit the original SAM interface, which is non-differentiable.
>
> SSC + MasP is not a simple modular combination, nor have existing works studied such components. The dynamic layer selection in SSC allows similarity maps from intermediate layers to directly participate in SAM mask generation. Thus, gradients from SAM can flow through the similarity maps back into the intermediate layers, which enhances the model’s focus on target regions, as evidenced by clearer activation cues (Figs. 4–6; Lines 840–930).
>
> **Notes:**
> >SSC is a reinforcement learning policy that, via stochastic sampling, allows each [SEG] token to slide across unrolled transformer layers, enabling dynamic layer selection at which it connects to the vision model (e.g., SAM) in a skip-connection fashion ("skip-connection-like''). By skipping subsequent ones of $\ell^*$, we can ''cheat'' this telephone game by letting the final participant (SAM) tap into intermediate layers (e.g., 1–39) in advance. Across several forward passes, such mechanism virtually connects all layers to SAM while activating only one pass at a time ("dropout-like''). Given the selected hidden layer, MasP uses the similarity map derived from the [SEG] token and image tokens as a soft logit mask to prompt SAM for mask generation, offering explicit spatial cues through its activation regions.
>
> ---
> **AW4:** (1) We do report model parameters and inference speed (Appendix B;Lines 640-657). (2) We have included visual comparisons with other methods [**[Links]**](https://github.com/anonymouslly/icml). (3) We do offer comparative validation of the key innovations w.r.t. SSC (Fig.2; Figs.4-6) + MasP (Table 2)  in Sec.3.2.

---

> > ### Author Rebuttal · Reviewer_DbMw · 2026-04-01
> >
> > The rebuttal has addressed my concerns, so I decide to raise my score.

---

### Official Review · Reviewer_rzFQ · 2026-03-11

**Soundness:** 3
**Presentation:** 3
**Significance:** 3
**Originality:** 3
**Overall Recommendation:** 5
**Confidence:** 4

**Summary:**

The paper proposes a unified visual grounding framework that can handle attribute variation in the visual grounding task by introducing PPM.
It addresses two limitations in the previous paradigm: (1) use hidden states from the fixed hidden layer, which may suffer from propagation errors; (2) only use \<SEG\> as a prompt, which ignores the explicit spatial cues.
Experiments on different datasets show the effectiveness of the proposed method.

**Compliance With Llm Reviewing Policy:**

Affirmed.

**Final Justification:**

The rebuttal clarifies my concern. I confirm my original score.

**Key Questions For Authors:**

Which layers are selected most frequently ? Whether the layer-selection distribution is stable across datasets or query types?

**Limitations:**

yes

**Strengths And Weaknesses:**

**strengths**

Overall, the figures and tables are clear.
The analysis is reasonable and easy to follow. Experiments are extensive and can show that the proposed method provide benifits.

**weakness**
1. The method is mainly about the prompting strategy between LLaVA and SAM (i.e.,  dynamically select a layer to get hidden states; use the semantic map apart from the seg token). The UGround model has no architectural changes. Why does the author claim it is a unified paradigm?

---

> ### Author Rebuttal · Authors · 2026-03-31
>
> **AW2:** As shown in Fig. 9 (Lines 1045–1070; see also [**[code]**](https://openreview.net/attachment?id=AXtUl2KiQU&name=supplementary_material)), UGround unifies LISA, SESAME, GSVA, and PixelLM into a single framework, thereby consolidating the tasks in Table 1 (Lines 30-40) under a unified pipeline.  Given that these models were originally designed for specific task settings, they achieve SOTA performance on their primary tasks, but are often ill-suited for others or leave certain tasks unaddressed. As shown in Table 1, tasks are scattered across different models with only partial coverage; moreover, implementation differences across models have led to the absence of a unified framework for integrating these tasks into a single pipeline.  A key challenge lies in unifying the interface of [SEG]/[REJ] tokens. e.g., in handling no-target cases, SESAME uses customized templates to reject queries with no target and correct them when a target exists [**[Links]**](https://github.com/anonymouslly/icml/blob/main/figs/f.png), whereas GSVA appends a [REJ] token but lacks such correction capability [**[Links]**](https://github.com/anonymouslly/icml/blob/main/figs/e.png).  Technically, integrating the tasks in Table 1 into a single pipeline requires unifying both dataloader interfaces and evaluation protocols.
>
> **We do construct a unified framework** (Table 1; see also: [**[Links]**](https://github.com/anonymouslly/icml)), but from an attribute perspective (Lines 42-53; 126-133), which diverges from prior works that primarily focus on versatile capabilities, such as performing segmentation and captioning jointly (e.g., OMG-LLaVA, GLaMM), or grounding across both images and videos (e.g., HyperSeg, OMG-Seg). This difference lies in the granularity of task settings when defining "unified."  The attribute perspective is novel. As emerging visual grounding tasks continue to expand the capability boundaries of LLMs, it is crucial to clarify whether such extensions occur at the macro level (versatility) or the micro level (attribute), which in turn facilitates a more mature visual grounding taxonomy.
>
> Our prompting strategy (SSC+MasP) does not directly contribute to a unified paradigm; however, as a plug-and-play module (i.e., requiring no architectural changes), it can be seamlessly incorporated into the UGround pipeline and enhance performance across various visual grounding tasks.
>
> Taken together, we believe that a) the unified UGround pipeline and b) the prompting strategy justify our contributions.
>
> Table 1: Attribute variation with respect to visual grounding. RES: Referring Expression Segmentation; RS: Reasoning Segmentation; FP: False Premise (reject queries with no target; **correct if target exists**); GRES: Generalized RES; Multi-RS: Multi-target RS.
>
> | Method | RES | RS | FP-RES | GRES | Multi-RS |
> |---|:---:|:---:|:---:|:---:|:---:|
> | LISA [CVPR'24] | ✓ | ✓ | ✗ | ✗ | ✗ |
> | SESAME [CVPR'24] | ✓ | ✓ | ✓ | ✗ | ✗ |
> | READ [CVPR'25] | ✓ | ✓ | ✓ | ✗ | ✗ |
> | GLaMM [CVPR'24] | ✓ | ✓ | ✗ | ✓ | ✗ |
> | GSVA [CVPR'24] | ✓ | ✓ | ✓ | ✓ | ✗ |
> | PixelLM [CVPR'24] | ✓ | ✓ | ✗ | ✓ | ✓ |
> | UGround (Ours) | ✓ | ✓ | ✓ | ✓ | ✓ |
>
> ---
> **AQ1:**  We report the results on the ReasonSeg test set, consisting of 799 samples, using LLaVA 7B (32 layers). Across short, long, and overall query settings, UGround shows limited sensitivity to query type. Although long queries exhibit slightly higher entropy, over 89% of selections fall within the top three layers (25, 31, 32) overall.
>
> | Metric               |   Short |     Long |   Overall |
> | -------------------- | :------: | :-------: | :--------: |
> | **Entropy**          |    1.39 | 1.52 | 1.46 |
> | **Gini**             |    0.64 | 0.67 | 0.66 |
> | **Top-3 (25/31/32)** |   86.9% | 91% |  89% |
>
> >**Entropy** measures the uncertainty of the layer selection distribution. ↓ lower → more concentrated (stable), ↑ higher → more diverse (dynamic).
> >
> >**Gini coefficient** quantifies the inequality of the distribution. ↓ lower → more concentrated on a few layers, ↑ higher → more dispersed across layers.
> >
> >**Top-3 (25/31/32)** represents the proportion of samples whose selected layers fall within the three most frequently chosen high-level layers.

---

> > ### Author Rebuttal · Reviewer_rzFQ · 2026-04-01
> >
> > The rebuttal clarifies my concern. I confirm my original score.

---

### Official Review · Reviewer_AyUQ · 2026-03-12

**Soundness:** 3
**Presentation:** 2
**Significance:** 3
**Originality:** 4
**Overall Recommendation:** 5
**Confidence:** 3

**Summary:**

This paper proposed UGround, a Unified visual Grounding paradigm that dynamically selects intermediate layers across Unrolled transformers as “mask as prompt”. In each forward, SSC enables the intermediate layer to connect with SAM, with MasP using the similarity map between image tokens and <SEG> token as a soft logic mask. The prompted mask is explicitly supervised. The method unifies the attribute-oriented segmentation, from explicit expressions to implicit instructions, one target to many targets, and positive queries to false premises within a single framework.

**Compliance With Llm Reviewing Policy:**

Affirmed.

**Final Justification:**

I would like to raise the rating to 5.

**Key Questions For Authors:**

- In my understanding, multiple forward passes (1 to T) select different intermediate layers and output the similarity map in the corresponding layer as the mask; then, in the training, are the losses for all passes summed together and backward?
- How to produce the inference with the multiple forward passes and the maps in selected layers?
- About the similarity part in Sec 3.2, corresponding to the texts that focus on the comparison in rows, the way of highlighting (bolding the highest in the column) is confusing.
- How to explain adding SSC to only Mprompt gets the results worse in Table 6 (Exp. 2 vs. Exp. 5)?
- In the ablation study in Tab. 7, there’s lacking the item for not training M explicitly. I'm wondering how it performs when using Mprompt while not forcing the similarity map to be like GT.

**Limitations:**

Yes

**Strengths And Weaknesses:**

Strengths:
- The proposed method is technically sound by clarifying the idea with analysis on the two key parts: layer selection and similarity map as a prompt.
- The experiments compared with sufficient related works also show that the method obtains superior performances.
- Dynamic selection of intermediate layers is novel compared with previous methods that only use the final layer token.


Weakness:
- The presentation about the method is not clear enough and is kind of hard to follow. Can see the concrete questions below.
- One contribution by the authors is the “unified” visual grounding from an attribute perspective. However, I didn’t see why the previous methods can not do that, and which proposed technique in this paper corresponds to enabling the “unified”.

---

> ### Author Rebuttal · Authors · 2026-03-31
>
> **AW2:** As shown in Fig. 9 (Lines 1045–1070; see also [**[code]**](https://openreview.net/attachment?id=AXtUl2KiQU&name=supplementary_material)), UGround unifies LISA, SESAME, GSVA, and PixelLM into a single framework, thereby consolidating the tasks in Table 1 (Lines 30-40) under a unified pipeline.  Given that these models were originally designed for specific task settings, they achieve SOTA performance on their primary tasks, but are often ill-suited for others or leave certain tasks unaddressed. As shown in Table 1, tasks are scattered across different models with only partial coverage; moreover, implementation differences across models have led to the absence of a unified framework for integrating these tasks into a single pipeline.  A key challenge lies in unifying the interface of [SEG]/[REJ] tokens. e.g., in handling no-target cases, SESAME uses customized templates to reject queries with no target and correct them when a target exists [**[Links]**](https://github.com/anonymouslly/icml/blob/main/figs/f.png), whereas GSVA appends a [REJ] token but lacks such correction capability [**[Links]**](https://github.com/anonymouslly/icml/blob/main/figs/e.png).  Technically, integrating the tasks in Table 1 into a single pipeline requires unifying both dataloader interfaces and evaluation protocols.
>
> **We do construct a unified framework** (Table 1; see also: [**[Links]**](https://github.com/anonymouslly/icml)), but from an attribute perspective (Lines 42-53; 126-133), which diverges from prior works that primarily focus on versatile capabilities, such as performing segmentation and captioning jointly (e.g., OMG-LLaVA, GLaMM), or grounding across both images and videos (e.g., HyperSeg, OMG-Seg). This difference lies in the granularity of task settings when defining "unified."  The attribute perspective is novel. As emerging visual grounding tasks continue to expand the capability boundaries of LLMs, it is crucial to clarify whether such extensions occur at the macro level (versatility) or the micro level (attribute), which in turn facilitates a more mature visual grounding taxonomy.
>
> Our prompting strategy (SSC+MasP) does not directly contribute to a unified paradigm; however, as a plug-and-play module (i.e., requiring no architectural changes), it can be seamlessly incorporated into the UGround pipeline and enhance performance across various visual grounding tasks.
>
> Taken together, we believe that a) the unified UGround pipeline and b) the prompting strategy justify our contributions.
>
> ---
> **AQ1-AQ2:** In original work LISA [1],
>
> input: ($x\_{img}$, $x\_{txt}$) $\rightarrow$ LLaVA (layers: $\ell=0,1,..., 31$) $\rightarrow$ **c) fixed last layer:** $\boldsymbol{h}\_{seg}^{\left( 31 \right)}\in \mathbb{R} ^{4096}$ $\rightarrow$ SAM: $\hat{\mathbf{M}}$
>
> In our UGround, we advance LISA  via components **a) Stochastic Skip Connection (SSC)** and **b) Mask
> as Prompt (MasP)**:
>
> input: ($x\_{img}$, $x\_{txt}$) $\rightarrow$ LLaVA (layers $\ell=0,1,..., 31$) $\rightarrow$ **(a) layer select**: e.g., $\ell^*$=18 $\rightarrow$ **(b) MasP**:$\mathcal{M}^{(18)}$, $\boldsymbol{h}\_{seg}^{\left( 18 \right)}\in \mathbb{R} ^{4096}$ $\rightarrow$ SAM: $\hat{\mathbf{M}}$
>
> (a) + (b) correspond to a single forward pass. Unlike LISA, which always uses the last layer, UGround dynamically selects the layer. This allows the model to skip subsequent layers (e.g., layers 18–31) and directly connect to SAM, which is why we refer to it as a "skip connection." The losses from all passes (1 to T) are not summed together; instead, they are sampled across different training iterations, rather than being computed jointly within a single iteration. This process can be analogized to dropout, where each forward pass activates only a subset of layers (i.e., truncated computation paths) during training. During inference, we use the hidden state from the last layer to fully exploit the model’s capacity. This is motivated by the observation that LLaVA’s layers are progressively aligned, with higher layers better capturing language-level semantics.
>
> ---
> **AQ3:** We have revised the table to improve clarity by removing column-wise highlighting.
>
> ---
> **AQ4:** Given that $\mathcal{M}_{prompt}$ is generated from the interaction between [SEG] token and image tokens ([SEG] token $\times$ image tokens), Exp. 2 removes the semantic guidance from [SEG] token when prompting SAM, which potentially weakens the representation capability of $\mathcal{M}\_{prompt}$. In contrast, comparing Exp. 5 to Exp. 6, when [SEG] token is introduced, the cIoU improves significantly from 52.42\% to 72.07\%.
>
> ---
> **AQ5:** We have added the item (Row 0) for not training M explicitly in Table 7:
>
> |Exp. ID|BCE|Dice|Gaus. BCE|Gaus. Dice|gIoU|cIoU|
> |-------|:-:|:--:|:----------:|:-----------:|:-------:|:-------:|
> | 0|||||65.82|69.23|
> ||||||||
> |1|✓||||64.56|71.44|
> |2||✓|||64.57|68.63|
> |3|||✓||63.16|65.74|
> |4||||✓|65.80| **73.49**|
> |5|✓|✓|||63.92|68.55|
> |6|||✓|✓|**66.13**|72.07|

---

> > ### Author Rebuttal · Reviewer_AyUQ · 2026-04-07
> >
> > Thanks for the responses. My concerns are basically addressed.

---

### Decision · Program_Chairs · 2026-04-30

**Decision:**

Accept (regular)

**Comment:**

he reviewers were unanimous in recommending acceptance, recognizing the paper as a technically solid and well-evaluated contribution with clear empirical improvements. They particularly appreciated the proposed unified visual grounding framework, the effective use of intermediate-layer selection and prompting, and the breadth of experiments across multiple grounding settings. While some reviewers initially raised questions about the clarity of the “unified” claim, the novelty relative to prior work, and several presentation details, the rebuttal addressed these concerns satisfactorily and helped clarify the paper’s motivation and scope. Overall, AC agrees with the reviewers that this is a strong submission whose strengths outweigh the remaining minor issues, and recommend acceptance.